# Presenting a comprehensive multi-scale evaluation framework for participatory modelling programs: A scoping review

Grace Yeeun Lee[1]*, Ian Bernard Hickie[1], Jo-An Occhipinti[1,2], Yun Ju Christine Song[1], Adam Skinner[1], Salvador Camacho[3,4], Kenny Lawson[1], Adriane Martin Hilber[3,4], Louise Freebairn[1,2,5]

1 Brain and Mind Centre, The University of Sydney, Sydney, NSW, Australia, 2 Computer Simulation & Advanced Research Technologies (CSART), Sydney, NSW, Australia, 3 Swiss Centre for International Health, Swiss Tropical and Public Health Institute, Basel, Switzerland, 4 University of Basel, Basel, Switzerland, 5 Research School of Population Health, The Australian National University, Canberra, ACT, Australia

☯ These authors contributed equally to this work.
* grace.lee@sydney.edu.au

**Data Availability Statement:** All relevant data are within the paper and its Supporting Information files.

## Abstract

### Introduction

Systems modelling and simulation can improve understanding of complex systems to support decision making, better managing system challenges. Advances in technology have facilitated accessibility of modelling by diverse stakeholders, allowing them to engage with and contribute to the development of systems models (participatory modelling). However, despite its increasing applications across a range of disciplines, there is a growing need to improve evaluation efforts to effectively report on the quality, importance, and value of participatory modelling. This paper aims to identify and assess evaluation frameworks, criteria, and/or processes, as well as to synthesize the findings into a comprehensive multi-scale framework for participatory modelling programs.

### Materials and methods

A scoping review approach was utilized, which involved a systematic literature search via Scopus in consultation with experts to identify and appraise records that described an evaluation framework, criteria, and/or process in the context of participatory modelling. This scoping review is registered with the Open Science Framework.

### Results

The review identified 11 studies, which varied in evaluation purposes, terminologies, levels of examination, and time points. The review of studies highlighted areas of overlap and opportunities for further development, which prompted the development of a comprehensive multi-scale evaluation framework to assess participatory modelling programs across disciplines and systems modelling methods. The framework consists of four categories

**Funding:** This research is being conducted under the Brain and Mind Centre's Right care, first time, where you live Program, enabled by a AUD12.8 million partnership with BHP Foundation. The Program will develop infrastructure to support decisions relating to advanced mental health, and guide investments and actions to foster the mental health and wellbeing of young people in their communities. The funders had no role in study design, data collection and analysis, decision to publish, or preparation of the manuscript.

**Competing interests:** Dr Louise Freebairn is currently employed part-time by the Brain and Mind Centre, University of Sydney; ACT Health as Director of Knowledge Translation and Health Outcomes, Epidemiology Section, ACT Health & CSART as Director of Policy Applications & Translational Science. A/Professor Jo-An Occhipinti is both Head of Systems Modelling, Simulation & Data Science at the University of Sydney's Brain and Mind Centre and Managing Director of Computer Simulation & Advanced Research Technologies (CSART). Professor Ian Hickie is the Co-Director, Health and Policy at the Brain and Mind Centre (BMC) University of Sydney. The BMC operates an early-intervention youth services at Camperdown under contract to headspace. He is the Chief Scientific Advisor to, and a 5% equity shareholder in, InnoWell Pty Ltd. InnoWell was formed by the University of Sydney (45% equity) and PwC (Australia; 45% equity) to deliver the $30 M Australian Government-funded Project Synergy (2017-20; a three-year program for the transformation of mental health services) and to lead transformation of mental health services internationally through the use of innovative technologies.

(*Feasibility*, *Value*, *Change/Action*, *Sustainability*) with 30 evaluation criteria, broken down across project-, individual-, group- and system-level impacts.

## Discussion & conclusion

The presented novel framework brings together a significant knowledge base into a flexible, cross-sectoral evaluation effort that considers the whole participatory modelling process. Developed through the rigorous synthesis of multidisciplinary expertise from existing studies, the application of the framework can provide the opportunity to understand practical future implications such as which aspects are particularly important for policy decisions, community learning, and the ongoing improvement of participatory modelling methods.

## Introduction

### Traditional *versus* participatory modelling

Systems modelling and simulation, also known as dynamic simulation modelling, is a term given to complex systems science analytic methods–such as system dynamics, Bayesian networks, and agent-based models–that is utilized in many countries and across diverse sectors to support evidence-informed decision making and to drive policy reform [1, 2]. By taking a complex systems view, significant challenges in society including population health crises, climate change, poverty, and civil strife can be better understood and managed through computer simulation models that capture the causal structure underlying the dynamics of these systems [1, 3–8]. Various systems modelling and simulation techniques have traditionally been applied across a range of disciplines including engineering, business, and environmental sciences for decades [9], but is now increasingly utilized in other fields including in public health [10–12]. This is largely attributed to the utility of systems modelling and simulation providing decision makers with both immediate and long-term support in understanding the prospective impacts of alternative strategic actions, where traditional statistical methods may be limited [13–16].

Systems modelling and simulation can provide insights at different levels of scale, including macro, meso, and micro; providing national, state, and local governments with tools that support strategic planning and decision making [4, 17–20]. By recognizing the interdependencies within complex systems, diverse stakeholder groups including those who are a part of the system being modelled are viewed as important communication agents. The involvement of stakeholders is necessary not only for their knowledge contributions but also their key role in coordinating the implementation of strategic system improvements–hence the value in shifting scientists away from working in isolation to develop systems models [21].

Participatory modelling (PM), or stakeholder-based systems modelling, brings together diverse knowledge and interests to engage in a joint learning and planning process to better understand complex systems, as well as possible implications of decision making to manage system challenges. Advances in technology and software have facilitated accessibility of modelling by a broader group of participants, allowing more diverse stakeholders working across a complex system to engage with and contribute to the development of these models [11, 22], as well as to inform decision making and further actions [23]. For example, graphical model interfaces allow stakeholders to better visualize and understand the logic and assumptions of a model than earlier software that required articulation of a model using mathematical equations

or computer coding. Such accessibility has also facilitated the participation of those most impacted by policy changes (such as consumer representatives)–helping to work towards all stakeholders sharing a common understanding of a complex problem or issue, inform and enhance collective action, assist collective decision making processes, enhance both individual and social learning, as well as precipitate changes in stakeholder behaviors [9, 21, 23–29].

## Evaluating participatory modelling programs–Challenges and opportunities

The inclusion of stakeholders during the PM process can facilitate learning, consensus, and transparent decision making [21]. However, PM evaluation is frequently disregarded or not based on transparent systematic methodological approaches [30]. A recent review paper of 60 randomly selected case studies on environmental PM programs reported that most studies (>60%) did not include evaluation [31]. The studies that did include evaluation were poorly reported, lacking detailed description and justifications on assessment criteria, methods of data collection, and analysis [31].

At the most basic level, evaluations provide systematic comparisons of program objectives and outcomes to understand how well something is working for the purpose of policy, planning, or implementation [32, 33]. According to the Cambridge Dictionary, evaluation is defined as the *"process of judging the quality, importance, amount, or value of something"* [34]. Applying this definition to the context of this paper, there is opportunity to better understand the quality, importance, and value of PMs [35]. This shifts the focus from solely one aspect of PM to a more holistic consideration of the whole PM process (e.g., knowledge integration and learning, technical systems model development, participatory and integrated planning, etc), providing opportunity for further knowledge on which aspects of PM are particularly important for policy decisions and community learning, as well as the ongoing improvement of PM methods [36, 37].

Evaluators are relied upon to address questions on the effectiveness of investments in local, state, and national programs, as well as to better explain if observed outcomes were (or were not) as planned, and how unintended consequences can be addressed [38–41]. There may be various motivations for conducting an evaluation of PM programs including the desire to improve and share knowledge on good practice for PM, quantitatively and qualitatively report on project impacts, as well as to assess the value of PM for future work [24, 35]. Evaluations also keep the modellers, funders, and other stakeholders of interest held accountable for demonstrating outcomes, as well as to provide merit to the work being evaluated [24]. Thus, PM program evaluations can also support policy makers to make evidence-informed decisions in determining how much weight to give the program or model outputs [24].

Evaluations that comprehensively capture the complex (e.g., uncertain, dynamic) nature of PM can be difficult [42], as embedding participatory approaches in systems modelling and simulation creates several challenges [35]. For instance, the focus of PM outcomes is often still on the knowledge integration and learning process rather than the multi-value perspectives integrated within the participatory process used to develop the models [31]. This can lead to evaluation practices to over- or under-represent certain stakeholder groups' experiences [31]. Additionally, previous studies that have attempted to evaluate the benefits of PM have reported difficulty in the design of the evaluation process, as complex systems rarely have comparative controls that allow for feasible experimental design (i.e., 'with modelling' intervention *vs* 'without modelling' control), making the measure of PM effectiveness challenging [35]. Thus, it is important to understand the distinction between the evaluation in detecting the effectiveness of the model development process, compared to the actual success or failure of the engagement with the model itself.

PM evaluations are also typically constrained by contextual factors including limited time and program budget. PM programs are often funded to the point of delivering the final model, rather than evaluating the process and benefits of PM including the extent to which the final model informed decision making or built consensus. A lack of investment in the evaluation of PM leads to decreased motivation to conduct thorough evaluations and may also risk evaluation efforts to be overly simplified when measuring the impact of PM [24], missing the opportunity to assess the performance of PM in different contexts to inform the adaption and improvement of processes [43].

### Objectives

Therefore, this scoping review aims to:

1. Identify published PM evaluation frameworks, criteria, and/or processes irrespective of the modelling method, or the discipline in which they are designed and/or implemented.

2. Assess the identified evaluation frameworks, criteria, and/or processes to understand their applicability to different PM program objectives and contexts.

3. Synthesize the findings to develop a novel evaluation framework that can be adapted and executed broadly across diverse PM programs, regardless of the discipline or modelling method. A flexible framework is necessary as PM itself requires flexibility to the potentially changing priorities and needs of participants.

A scoping review has been deemed the most appropriate approach, compared to a systematic review, as the purpose of this paper is to focus on the broad collection and discussion of available literature, and to present a comprehensive multi-scale evaluation framework for PM programs [44, 45]. Scoping reviews are better suited than systematic reviews when the aim of evidence synthesis is to provide an overview of literature and identify broad knowledge gaps in a topic that has not been extensively reviewed (as opposed to seeking to answer focused questions as is done in systematic reviews) [44, 45]. Scoping reviews also differ from non-systematic literature searches as they are routinely informed by an a priori protocol, and are conducted via a rigorous and transparent approach to minimize error as well as to ensure reproducibility [44]. The development and application of the presented evaluation framework is supported by a participatory systems modelling program for youth mental health (described below in the *Discussion*). To our knowledge, this is the first multidisciplinary scoping review of evaluation frameworks for PM programs.

### Materials and methods

This scoping review was conducted according to the suggested methodology outlined in the *Joanna Briggs Institute (JBI) Reviewers' Manual for Evidence Synthesis* [46], in combination with additional recommendations for conducting scoping reviews [47]. The Preferred Reporting Items for Systematic Reviews and Meta-Analyses (PRISMA) statement was also applied [48], and the PRISMA extension for Scoping Reviews (PRISMA-ScR) Checklist has been provided as Supporting Information (S1 File). This review paper has also been registered with the Open Science Framework [49].

### Search strategy

A focused search was conducted via Scopus in May 2021 in consultation with an academic librarian at The University of Sydney, utilizing a combination of Boolean operators, wildcards, and truncations to develop the final search strategy (Table 1). Scopus is a meta-database, and

**Table 1. Search strategy.**

| Topic | Searches | Results |
|---|---|---|
| **Scopus** | | |
| (Participatory) Systems modelling and simulation | (TITLE-ABS-KEY ("simulation model*")) OR (TITLE-ABS-KEY ("system* model*")) OR (TITLE-ABS-KEY ("participator* model*")) OR (TITLE-ABS-KEY ("system* dynamic*")) OR (TITLE-ABS-KEY ("agent-based model*")) OR (TITLE-ABS-KEY ("discrete event simulation")) OR (TITLE-ABS-KEY ("Bayesian network*")) OR (TITLE-ABS-KEY ("hybrid simulation")) OR (TITLE-ABS-KEY ("system* science")) OR (TITLE-ABS-KEY ("stakeholder-based model*")) | 256,087 |
| Participatory design | (TITLE-ABS-KEY ("participator*")) OR (TITLE-ABS-KEY (co-design*)) | 89,247 |
| Evaluation | (TITLE-ABS-KEY (evaluat*)) OR (TITLE-ABS-KEY ("review*")) | 13,380,297 |
| Combined queries | | 465 |
| **Additional search** | | |
| Hand searching, co-author recommendations, citation chaining | | 10 |
| **Total yielded literature** | | **475** |

Search terms were selected to ensure the literature will provide a focused yield on the topic. For example, terms such as "co-design" and "participatory model" were selected in favour for terms such as "co-creation" and "co-production" as these additional terms yielded literature that were not relevant to the focus topic.

includes records from various databases across disciplines including environmental sciences, engineering, mathematics, social sciences and medicine [36]. Additional searches were conducted through hand searching, co-author recommendations, and citation chaining.

## Inclusion and exclusion criteria

The criteria for inclusion were defined a priori by the authors (GYL, LF) in a Population, Concept, Context format [46], and applied to all yielded records. As detailed in Table 2, this scoping review included sources that described an evaluation framework, criteria, and/or process for PMs. Though there are varying definitions that exist, for the purpose of this review, we have defined an *evaluation framework* as a tool that presents an overview of the evaluation theory, topics or themes, questions, and/or data sources; *evaluation criteria* as a performance metric or indicator that further breaks down the evaluation framework, and; *evaluation process* as a defined procedure guided by theory of how the authors recommend PM evaluation [50].

Records that presented a standalone theoretical framework (or applied via a case study example) were also included. In contrast, records that only described the methodological tools

**Table 2. Inclusion and exclusion criteria in a population, concept, context format, as recommended by JBI.**

| | Inclusion criteria | Exclusion criteria |
|---|---|---|
| **Population** | Not defined (due to the limited number of PM evaluation frameworks, criteria, and/or processes, broadened the 'population' category to not be defined to specific fields of disciplines/population groups). | |
| **Concept (e.g., PM program methods)** | Records describing an evaluation framework, criteria, and/or process to support the evaluation of a PM program (standalone theoretical framework or applied in a case study). | Records solely describing the methods adopted to evaluate the implementation of PM programs, without describing an evaluation framework, criteria, and/or process. Records solely describing the evaluation of the technical model (i.e., not PM). Records describing PM implementation programs not evaluated. |
| **Context (e.g., country, setting)** | Not defined (due to the limited number of PM evaluation frameworks, criteria, and/or processes, broadened the 'context' category to not limit any cultural, geographic, or specific setting factors). | Outcomes not published in English. |

(e.g., interviews, etc) used to evaluate the implementation of PM programs without describing an evaluation framework, criteria, and/or process were excluded. Records that only described the evaluation of a technical model (i.e., not PM) were excluded, as were records that described PM implementation programs without any consideration of evaluations. Date limits were not set, but studies not published in English were excluded from the review.

## Data extraction and synthesis

Using a pro forma approach, the first author (GYL) independently reviewed the titles and abstracts of all yielded records. Uncertainty regarding whether records met the inclusion criteria were resolved through two-weekly discussions with the senior author (LF). To verify the data extraction, a random sample of 10 records were independently checked by LF. Following this verification process, full text review and data extraction was conducted independently by GYL.

To address the first and second objective, a data extraction template was developed by the authors (GYL, LF). Data extraction templates are used in scoping reviews to provide a structured and detailed summary of each record [46], and were used to collate information on yielded records that underwent full text review. The four-dimensional framework (4P) developed by Gray *et al.* formed the basis of the data extraction template. The 4P framework was selected as it was developed specifically to standardize the reporting of PM programs and therefore provided a useful structure to guide data extraction [37]. This framework has since been adapted to include two additional dimensions by Freebairn–imPact and Prioritizing [13]. The definitions of the resulting six dimensions (6P) were slightly adapted to fit the evaluation objectives of this scoping review. The revised definitions are: *Purpose* (why PM approaches should be evaluated); *Process* (the method utilized to execute evaluation framework/criteria/process); *Partnerships* (which stakeholders were involved in the development of the evaluation framework/criteria/process); *Products* (evaluation approach–e.g., theoretical, conceptual, and/or implementation); *imPact* (what were the outcomes/strengths of the evaluation framework/criteria/process), and; *Prioritizing* (what were the barriers/future opportunities of the evaluation framework/criteria/process) [13]. To ensure an all-inclusive synthesis of records, the JBI template for data extraction [46], as well as additional elements included by the authors were also incorporated into the final data extraction template (Table 3). Once the author (GYL) completed full text review, the senior author (LF) reviewed and verified the final list of records to include for synthesis.

To address the third objective, a narrative synthesis of the findings was conducted and utilized to develop an evaluation framework that can be applied across diverse disciplines and modelling methods. A narrative synthesis allows for the in-depth exploration beyond the description of the included records to understand relationships (e.g., differences, similarities) between the studies [51].

As part of the narrative synthesis, a word cloud was generated to analyze the heterogeneity in terminology identified during the data extraction process for the studies included in the scoping review [52]. Word clouds visually display the most frequently used words in a body of text with the bigger font size of a word illustrating that this word is used more frequently [53]. To ensure that a focused word cloud was generated specific to evaluation, the full text of all 11 studies included for synthesis were uploaded, which was followed by a process of elimination whereby words that were not related to evaluation–such as *university*, *platform*, and various stop words–were deleted. Following this process, the authors went through the remaining list of words and merged synonyms as well as the same words presented in its singular or plural form or with tense variation.

Though word clouds can support the identification of commonly utilized terms, there are limitations. For example, it is not clear from just the word cloud exercise alone how many

**Table 3. Data extraction template.**

| Source* | Data to be extracted |
|---|---|
| JBI | Reference |
| | Author, year of publication, title, journal, volume, issue, pages |
| JBI | Population |
| JBI | Concept (e.g., PM program methods) |
| JBI | Context (e.g., country, setting) |
| 6P | Purpose (e.g., why PM approaches should be evaluated) |
| 6P | Process (e.g., method utilized to execute evaluation framework/criteria/process) |
| 6P | Partnerships (e.g., stakeholders involved in the development of the evaluation framework/criteria/process) |
| 6P | Products (e.g., evaluation approach–e.g., theoretical, conceptual, implementation) |
| 6P | imPact (e.g., outcomes/strengths of the evaluation framework/criteria/process) |
| 6P | Prioritizing (e.g., barriers/future opportunities of the evaluation framework/criteria/process) |
| GYL, LF | Other important themes? |
| GYL, LF | Evaluation framework criteria |
| GYL, LF | Include in scoping review? |
| GYL, LF | Other comments |

*_JBI_ refers to the Joanna Briggs Institute's Reviewers' Manual for Evidence Synthesis; _6P_ refers to Freebairn's adapted framework to standardize reporting of PM programs; _GYL_, _LF_ refers to additional elements included by the first and senior authors to ensure an all-inclusive data extraction process.

terms appeared in each individual study. Additionally, some words may have different meanings depending on the field that the paper was published in (e.g., the word 'sustainability' may have a different meaning when used in environmental PM _vs_ in a public health PM program). Thus, the whole author group engaged in an iterative process whereby the word cloud was utilized as a discussion tool to provide feedback, refine, and finalize the presented evaluation framework. Discussions with the authors were facilitated by GYL and LF from April to July 2021 via informal and formal meetings.

## Results

### Part I: Scoping review

The initial Scopus search yielded 465 results, and an additional 10 records were identified through hand searching, co-author recommendations, and citation chaining. Most articles were excluded from review based on their titles and abstracts (n = 451), as the majority described the evaluation methods or outcomes of the implementation of a PM program without any reference to an evaluation framework, criteria, and/or process. After screening 24 full-text records, 11 studies were included for synthesis. Though it was not intentional, all included records were from academic journals, as opposed to grey literature and conference papers. The PRISMA flow diagram is presented in Fig 1.

### Characteristics of studies

Table 4 summarizes the characteristics of the studies. All but one (10/11, 90.9%) were published in an environmental sciences journal. The remaining one study was published in the

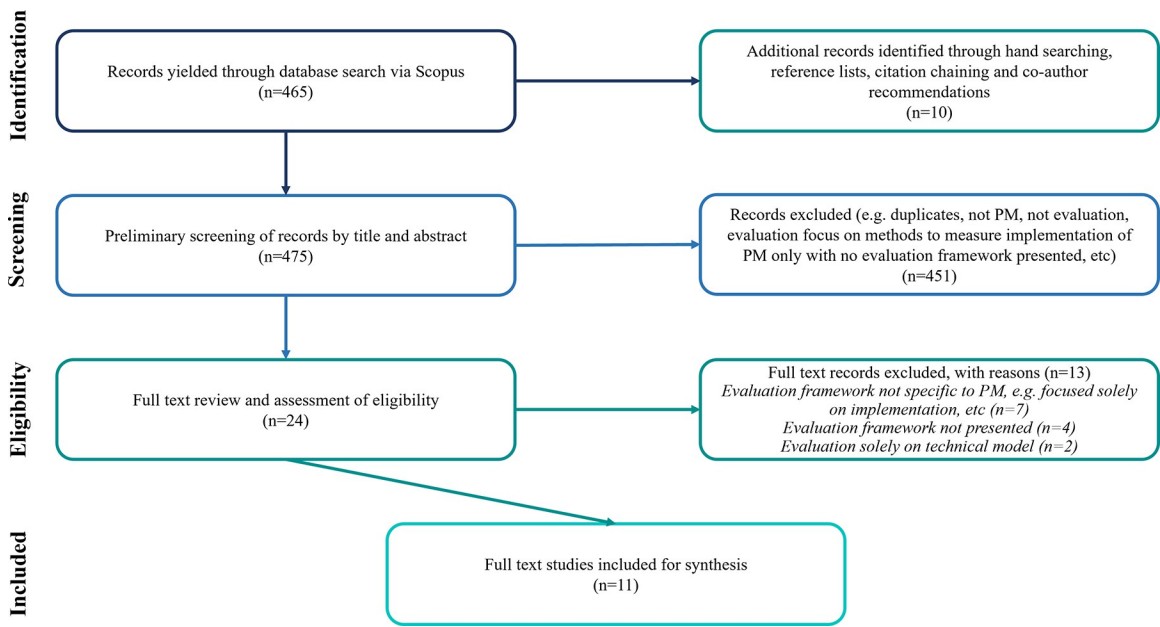

**Fig 1. PRISMA flow diagram.**

**Table 4. Characteristics of included studies.**

| First author | Year of publication | Name of article | Academic journal published |
|---|---|---|---|
| **Lynam** [54] | 2007 | A Review of Tools for Incorporating Community Knowledge, Preferences, and Values into Decision Making in Natural Resources Management | Ecology and Society |
| *Jones* [25] | 2009 | Evaluating Participatory Modelling: Developing a Framework for Cross-Case Analysis | Environmental Management |
| *Zorilla* [55] | 2010 | Evaluation of Bayesian Networks in Participatory Water Resources Management, Upper Guadiana Basin, Spain | Ecology and Society |
| *Matthews* [56] | 2011 | Raising the bar? The challenges of evaluating the outcomes of environmental modelling and software | Environmental Modelling & Software |
| *Smajgl* [57] | 2015 | Evaluating participatory research: Framework, methods and implementation results | Journal of Environmental Management |
| *Maskrey* [58] | 2016 | Participatory modelling for stakeholder involvement in the development of flood risk management intervention options | Environmental Modelling & Software |
| *Falconi* [35] | 2017 | An interdisciplinary framework for participatory modelling design and evaluation–What makes models effective participatory decision tools? | Water Resources Research |
| *Hedelin* [59] | 2017 | Participatory modelling for sustainable development: Key issues derived from five cases of natural resource and disaster risk management | Environmental Science and Policy |
| <u>Hamilton</u> [24] | 2019 | A framework for characterizing and evaluating the effectiveness of environmental modelling | Environmental Modelling & Software |
| *Waterlander* [60] | 2020 | A System Dynamics and Participatory Action Research Approach to Promote Healthy Living and a Healthy Weight among 10–14-Year-Old Adolescents in Amsterdam: The LIKE Programme | International Journal of Environmental Research and Public Health |
| *Zare* [61] | 2021 | A formative and self-reflective approach to monitoring and evaluation of interdisciplinary team research: An integrated water resource modelling application in Australia | Journal of Hydrology |

***Bold and italics = clear evaluation framework and/or criteria defined, with case study described;* Bold = clear evaluation framework and/or criteria defined, with no case study**; *italics = evaluation process defined, with case study described;* <u>*Italics and underline = clear evaluation framework, criteria and/or process defined, with no case study*</u>.

International Journal of Environmental Research and Public Health. Approximately half (7/ 11, 63.6%) of the included studies were published in the past decade (2011–2021). Overall, three quarters (346/475, 72.8%) of the 475 total yielded records reviewed by their abstracts and titles were published in the last decade.

**Characteristics of evaluation frameworks, criteria, and/or processes.** The papers included for synthesis either described a theoretical evaluation framework and/or criteria with no application to a case study [24, 54]; described a theoretical evaluation framework and/or criteria applied to a case study [25, 35, 55, 58, 59], or; described an evaluation process applied to a case study [56, 57, 60, 61]. The majority of the evaluation frameworks, criteria and/or processes were developed by building upon already existing work [24, 25, 35, 55–61]. For example, Cash *et al*'s paper–which described how policy makers were more likely to use scientific evidence if three criteria are met (i.e., credible, salient, legitimate) [62]–was utilized in Hamilton *et al*. and Falconi *et al*'s framework [24, 35]. Rowe and Frewer's work was cited by two different studies [63], with Maskrey *et al*. referencing this work to include as an evaluation criteria to assess accessibility (e.g., language) during the participatory process, whereas Zorilla *et al*. referenced it as an evaluation criteria to assess how the participatory process enabled stakeholder values, assumptions, and preferences to be incorporated into decision making [55, 58]. Two of the evaluation frameworks described an empirical process of how their frameworks were developed, supplemented with literature reviews [24, 54].

Key benefits and areas of future research were identified for each paper through the data extraction process (i.e., imPact and Prioritizing categories of Freebairn's adapted 6P communications framework on reporting PM outcomes). For example, Lynam *et al*.'s paper was one of the first academic papers that attempted to address the gap in research evidence to support improved evaluation practices in PM [54]. It was also one of the first to identify the need to address power relations when working with communities, as well as the PM process as distinct from the technical model (e.g., encouraging co-learning/communication *vs* level of accuracy/precision) [54]. As such, Lynam *et al*. was referenced by various other papers [25, 35, 55, 58, 59], and was used by Zorilla *et al*. to inform the development of their own evaluation framework [55]. While Lynam *et al*.'s work was ground-breaking, being one of the first in this field, it was limited in providing a comprehensive description of the theoretical underpinnings of their framework.

The evaluation frameworks, criteria, and/or processes focused on different design features and levels of examination (e.g., project *vs* system level, short *vs* long-term observation, etc). For example, Jones *et al*., Hamilton *et al*., Zare *et al*., and Smajgl & Ward noted the importance of evaluation both *ex ante* and *ex post* to ensure all involved–including modellers and stakeholders–have a better understanding of what the PM process is aiming to achieve, and to keep everyone accountable to the defined objectives [24, 25, 57, 61]. Jones *et al*. and Hamilton *et al*. also differentiated between the levels of stakeholder participation [24, 25]; and although this was not embedded into the evaluation criteria presented by Zorilla *et al*., there was recognition that future work should distinguish evaluation amongst stakeholder groups from policy makers to farmers [55]. There were two papers that did not explicitly consider the multi-value perspectives integrated within PM, such as consideration of the multiple levels of examination (e.g., diverse stakeholder perspectives, project *vs* system level, short *vs* long-term observation, etc) [54, 56]. Maskrey *et al*., Falconi *et al*., and Hamilton *et al*. recognized that evaluations should also consider both the immediate and long-term outcomes [24, 35, 58]. Hedelin *et al*. and Waterlander *et al*. focused on the organizational level–for example, focusing on the various elements of the system to understand organizational learning, change, and action [59, 60]. This information is summarized in Table 5.

A strength of some of the identified evaluation frameworks, criteria, and/or processes was that they were not only applicable to the specific PM program it was designed for, but

**Table 5. Characteristics of evaluation frameworks, criteria, and/or processes reported through individually included studies.**

| First author (Year) | Brief description of evaluation framework, criteria, and/or process | Method of evaluation framework, criteria, and/or process of development | Application of evaluation framework, criteria, and/or process | Level of examination (project *vs system* level, short *vs* long-term observation, etc) | Strengths and limitations |
|---|---|---|---|---|---|
| **Lynam (2007)** | Framework and criteria presented to evaluate various PM tools (e.g. Bayesian belief network, systems models, etc) in the natural resource management context. Three categories of evaluation criteria: 1) *Capabilities*, evaluating potential applications of the participatory tool, 2) *Use*, evaluating conditions or context of use, and 3) *Products*, evaluating the nature of the results/outputs of the tools. Full criteria presented in *Tables 2, 3* and *4*. | Not described at length, but a discussion-based process amongst authors, with cross-examination with the literature to validate discussions briefly described. | Authors have stated that the decision to not demonstrate the application of the evaluation framework with field examples was intentional, with the justification that the amount of text needed to provide adequate context for each PM tool was necessary. | N/A–Not explicitly mentioned | *Strengths.* One of the first academic papers that considered systematic evaluation approaches in PM, and considers both the process (e.g., co-learning), as well as the technical model (e.g., reasonable levels of accuracy). Enables others to adopt the most appropriate PM tool by considering strengths and limitations presented. *Limitations.* Limited description regarding the empirical evidence utilized to develop the presented evaluation framework and criteria. |
| *Jones (2009)* | Evaluation framework (named *Protocol of Canberra*) developed in the context of natural resource management to assess extent to which different PM initiatives not only modify perceptions, but also facilitate change in interactions between participants to make collective decisions. Consists of two main components: 1) *Designers Questionnaire*, which captures the project team's experiences, and 2) Participants *Evaluation Guide*, which captures the participants' experiences of the project. | Developed through a collaboration between French and Australian researchers in 2005 (ADD-ComMod project) engaged in PM and evaluation research. Framework designed to identify *context* (e.g., setting), process (e.g., method), and underlying theoretical thread (e.g., theory-based evaluation to understand link between theory and practice, building off work of Argyris [64], Patton[65], and Curnan) [66]. | Presented three case studies as part of the ADD-ComMod evaluation project, to demonstrate the use of the evaluation framework in different contexts: 1) *AtollGame*, ex post evaluation of groundwater management in the Republic of Kiribati, 2) *Catalyst Project*, ex post evaluation of strategic regional development planning conducted by CSIRO, and 3) *Lower Hawkesbury Estuary Management Plan*, ongoing process evaluation to manage the Lower Hawkesbury Estuary in Australia. | Level of evaluation differentiated by research participants *vs* project team. Can be applied as either an *ex post* (conducted after implementation) or ongoing evaluation (carried out during the project as it progresses). | *Strengths.* Evaluation framework is flexible to encompass various approaches to PM. Consideration of multiple stakeholder perspectives, at various evaluation time points. *Limitations*: The authors of the evaluation framework admitted that the robustness of their evaluation framework has come at the cost of simplification–specifically, assumption of linear structure of framework. |
| *Zorilla (2010)* | Process evaluation criteria for public participation (and its PM tools), with an emphasis on "what works best when" in the context of water resources management. Evaluation criteria broken down to the *participatory process* (e.g., improve system understanding, foster trust, etc), as well as *capabilities of Bayesian networks* (e.g., graphical interface, level of knowledge or uncertainty, etc). | Process evaluation followed methodology outlined by Abelson *et al*. (2003) [67], Rowe and Frewer (2004) [68], and Von Korff (2006) [69]. Criteria used to evaluate the PM process itself (e.g., increase stakeholder understanding, foster trust, etc), as well as to evaluate the quality of Bayesian networks as a PM tool (e.g., graphical interface, etc). References to each evaluation criteria provided (*Tables 3* and *4*); criteria to evaluate PM tool belongs to three categories described by Lynam *et al*., 2007 [54]–*capabilities*, *use* and *products*. | Evaluation criteria applied to evaluate the ability of Bayesian networks to effectively engage stakeholders and to support decision making in complex situations (challenges caused by uncontrolled groundwater extraction in Upper Guardiana Basin, Spain). Employed mixed methods: 1) *Stakeholder evaluation questionnaires*, 2) *Semi-structured telephone interviews*, 3) *Researchers' theoretical assessment* (Stewart *et al*., 1984 [70]; Einsidiel *et al*., 2001 [71]; Henriksen *et al*., 2007 [72]). | Not clear. However, authors have recognized that future studies should distinguish evaluation amongst stakeholder groups (policy makers *vs* technicians, farmers, and environmentalists). | *Strengths.* Presentation of evaluation criteria which considers participatory process, as well as influence of the actual model itself. Evaluation framework is built on research evidence, with application and outcomes exemplified through a case study. *Limitations.* Authors have recognized the short length of the questionnaire may have been a limitation (i.e., more rigorous development of evaluation tools are required). |

*(Continued)*

**Table 5.** (Continued)

| First author (Year) | Brief description of evaluation framework, criteria, and/or process | Method of evaluation framework, criteria, and/or process of development | Application of evaluation framework, criteria, and/or process | Level of examination (project vs system level, short vs long-term observation, etc) | Strengths and limitations |
|---|---|---|---|---|---|
| *Matthews (2011)* | Conceptual evaluation process described that situates outcome evaluation within the wider context of environmental modelling and software activity (EMS) to recognize the differentiation between *outcomes* (changes in values, attitudes and behavior) and *outputs* (knowledge mobilized in peer reviewed articles, software, or datasets). The conceptual evaluation process consists of three loosely coupled phases that link EMS research to outcomes–*research*, *development* and *operations*–in which evaluation plays an integral role across all phases. | Evaluation process built on understanding of the relationship between *context*, *process* and *outcomes* (Blackstock *et al.*, 2007 [73], and; Patton, 1998 [74]. Conceptual evaluation process is a generalization of the "consultancy model" for successful Decision Support System proposed by McCown (2002) [75], where knowledge is passed between phases rather than software tools. | Applied evaluation process in a case study (Communicating Climate Change Consequences for Land Use, C4LU project). Outcome evaluation conducted utilizing different methodological approaches depending on the phase (e.g., *research*–peer review and validation; *development*–parallel process of software engineering and quality assurance, as well as workshops with national policy makers, and; *operations*–evaluation form and workshop recordings. | N/A–Not explicitly mentioned. | *Strengths.* Recognition of the importance of conceptual processes to be coupled with practical evaluation methods. *Limitations.* Evaluation process is arguably too simple with no details on evaluation framework or criteria. Authors have disclosed that while a simple evaluation of the C4LU workshop outcomes was conducted, more work is required on the design and interpretation of such evaluation processes. |
| *Smajgl (2015)* | Evaluation protocol to facilitate systems learning through a structured participatory process by decision makers concerned with the management of environmental resources. | Evaluation protocol built off of the Challenge and Reconstruct Learning (ChaRL) Framework (Smajgl and Ward, 2013) [76], which draws on theoretical concept of complex systems, decision making constructs, and influences of learning (e.g., values, beliefs and attitudes). | Evaluation of series of PM workshops where an agent-based simulation model was used to challenge existing beliefs concerned with the effectiveness of proposed policy actions and development ideas in the Greater Mekong Subregion. Employed a mixed methods (i.e., questionnaires and workshops); decision maker learning was evaluated via observed changes in individually held values, beliefs, and attitudes. | Not clear; participation of workshops was targeted for stakeholders that had some degree of influence to make decisions. | *Strengths.* Recognition of the importance of both *ex ante* and *ex post* evaluation approach. *Limitations.* Theoretical underpinnings of the evaluation protocol rely heavily on decision making evidence, rather than considering evidence on how to effectively evaluate PM as a whole (both process and technical model). |
| ***Maskrey (2016)*** | Evaluation framework designed to understand the benefits and limitations of the PM process itself, and assessment of outcomes. Evaluation framework executed via *process evaluation* (criteria broken down broken down across themes: *accessibility*, *deliberation*, *representation*, *responsiveness*, *satisfaction)* as well as *outcome evaluation* (broken down into *substantive outcomes* and *social outcomes*). | Developed from synthesis of frameworks by Beierle (1999) [77], Rowe and Frewer (2000) [63], and Webler and Tuler (2002) [78]. Additional references provided for individual criterion in *Tables 7* and *9*. | Evaluation framework applied in a simple Bayesian network model to exemplify how PM can support local flood risk management contexts in Hebden Water catchment (UK). | Evaluation framework enables the consideration of the process and final outcome (e.g., short- vs long-term outcomes). | *Strengths.* Consideration of both short-term substantive and longer-term social benefits. Comprehensive evaluation criteria presented with clear disclosure of evidence base, as well as key findings from the Hebden Water catchment case study. *Limitations.* Though emphasis was made on short- vs long-term outcomes, not clear in case study whether sufficient time between evaluation points would have allowed for long-term outcomes reporting. |

*(Continued)*

**Table 5.** (Continued)

| First author (Year) | Brief description of evaluation framework, criteria, and/or process | Method of evaluation framework, criteria, and/or process of development | Application of evaluation framework, criteria, and/or process | Level of examination (project vs system level, short vs long-term observation, etc) | Strengths and limitations |
|---|---|---|---|---|---|
| *Falconi (2017)* | Two-stage PM evaluation framework which aimed to improve model effectiveness as participatory tools through standardizing data, documentation, and reporting practices. *1) Stage one*: Five dimensions of public participation in environmental decision making: *participants*, *stages of participation*, *degree of involvement*, *level of influence*, *purpose (who, when, how, why)*, and; *2) Stage two*: Attributes of successful PMs–categorized into three evaluation criteria: *credibility*, *salience*, *legitimacy*. | Stage one adapted from the National Resource Council (2008) [79]; Stage two expanded characteristics described by Cash *et al.*, 2003 [62], and Carr *et al.*, 2012 [80]. | Application of evaluation criteria across five distinct case studies: 1) *Community-based forest management*, Zimbabwe, 2) *Shared vision modelling for ACT-ACF water basin*, USA, 3) *Water management alternatives*, USA, 4) *Water resource allocation*, *Solomon Islands*, and 5) *Regional land-use*, Senegal River Delta. | Consideration of intermediate outcomes (distinct from 'ultimate outcomes'). | *Strengths*. Two-stage evaluation framework provides standardized mechanism that capture both the technical and social nature of PM. Flexibility to apply the framework in wide range of cases across disciplines. *Limitations*. By way of the authors demonstrating the applicability of their framework in five case studies, there was no comprehensive application of the framework (e.g., only narrative summaries described). |
| *Hedelin (2017)* | Evaluation framework presented with 16 criteria for procedural sustainable development, with key sustainability principles (*integration*, *participation*) applied across themes (*across disciplines; across values; contributing to the process; generating commitment, legitimacy or acceptance; across organizations*) | Development process not described; reader directed to Hedelin 2007 [81], Hedelin 2015 [82], and Hedelin 2016 [83] for more information on theoretical basis of the evaluation framework. | Framework applied to five case studies on large PM projects that span from water management to flood risk management programs. Did not report on individual case studies but rather, themes of case studies applied to the evaluation framework. | Not clear; however, evaluation criteria derived from research on organization learning, multilevel governance, organization coordination, and collaborative planning. | *Strengths*. Evaluation framework developed for sustainable development research, but can be applied to other disciplines of research. *Limitations*. Despite the emphasis on decision support, evaluation criteria of the five case studies did not detail decision support outcomes. |
| *Hamilton (2019)* | Comprehensive formative evaluation framework to consider multiple dimensions and perspectives in environmental PM studies. Evaluation criteria grouped into eight categories: *project efficiency*, *model accessibility*, *credibility*, *salience*, *legitimacy*, *satisfaction*, *application*, and *impact*. Four-step evaluation process also described: 1) *Identify project context affecting evaluation*, 2) *Identify evaluation context affecting method*, 3) *Design evaluation process based on the project and evaluation context*, and 4) *Execute evaluation plan and use learnings to improve current and future projects.* | Evaluation framework developed as part of workshop process, based on participants' understanding and supported by literature review. Participants not clearly identified, but reported to have diverse range of expertise including "...social and natural sciences, public health, and computer science [with] extensive experience in the development of models for decision and policy support, social learning, and scientific research." Built on work of others including but not limited to Cash *et al.*, 2003 [62], Goeller (1998) [84], and Roughley (2009) [85], with note that on their own, these frameworks are too generic and only relevant to environmental modelling processes. | Total of 32 evaluation criteria presented, with demonstration of how three common types of research methods–decision support systems, PM, and research modelling–should prioritize the criteria for evaluation purposes. Overview of common evaluation research methodologies presented in *Table 4*. | Levels of evaluation impacts differentiated, including *project-level* (within project timeframe), *group-* and *individual-level impacts* (apparent in short to long term), *system-level* (becoming apparent in long-term). Evaluation criteria and process to be applied both before (*ex ante*) and after (*ex post*) the PM program. | *Strengths*. Though evaluation criteria are specific to environmental modelling studies, can be adapted across diverse research disciplines. *Limitations*. Though 32 evaluation criteria are very comprehensive, it is not always pragmatic and realistic to achieve in all evaluation research studies. There is no research evidence of the actual effectiveness of the evaluation framework when applied in a PM program. Research tools are also not provided to guide the adoption of the framework in evaluation practice. |

*(Continued)*

**Table 5.** (Continued)

| First author (Year) | Brief description of evaluation framework, criteria, and/or process | Method of evaluation framework, criteria, and/or process of development | Application of evaluation framework, criteria, and/or process | Level of examination (project *vs* system level, short *vs* long-term observation, etc) | Strengths and limitations |
|---|---|---|---|---|---|
| *Waterlander (2020)* | Evaluation process designed to understand how the system evolves under influence of the LIKE programme, which aims to address the complex problem of childhood overweight and obesity in 10-14-year-old adolescents through PM. | Evaluation process built on work by Moore *et al.*, 2015 [86]; Walton (2015) [87], and; Egan *et al.*, 2019 [88]. Also developed underpinned by principles of Participatory Action Research and developmental evaluation. | Evaluation process will be applied in the LIKE programme employing qualitative and quantitative methods to assess changes in health behavior and body weight that result from the programme and interpret these outcomes in relation to the system changes. | Consideration of prospective evaluation data collection that measures pre-existing systems maps, with changes over time at different levels of the system elements, system structures, and/or the system as a whole. Similar to that proposed by Egan *et al.*, placed specific emphasis on achieving changes at higher system levels throughout the action-programme development using the Intervention Level Framework. | *Strengths.* Developmental systems evaluation design, built on principles of Participatory Action Research, leads to adaptive evaluation process that has the potential to respond to changes at different levels of the system. *Limitations.* No research evidence currently available regarding effectiveness of the evaluation process when applied in a PM program, such as LIKE. |
| *Zare (2021)* | Formative, self-reflective monitoring and evaluation process described consisting of six steps: *1) Review of documents to understand research goals, priorities and records, 2) Pathways assessment, 3) Reflective meetings with each team member, 4) Analysis of qualitative data from meetings to capture perceived strengths, weaknesses, success and failures, and suggested actions, objectives and goals, 5) Survey to elicit feedback on identified goals, objectives and actions*, and *6) Report of approved actions based on the conducted survey of team members.* | Monitoring and evaluation process developed as a result of the synthesis of methods from Gibbs (1988) [89], Holzer *et al.* (2018) [90], Kunseler *et al.* (2015) [91], van Mierlo *et al.* (2010) [92], Zare *et al.* (2020) [93]. | Evaluation process applied to an integrated assessment of water allocation and use opportunities modelling project in the Campaspe catchment, part of the Murray-Darling Basin in Victoria (Australia), to respond to challenges regarding water availability. | Recognition that monitoring and evaluation processes are an integral activity during all steps/phases of PM to aim for ambitious outcomes and modify activities over time, as needed. | *Strengths.* Adaptive and flexible monitoring and evaluation process to suit the needs of complex problem solving. *Limitations.* A general process is described, rather a comprehensive evaluation framework/ criteria. The authors have recognized that it would have been advantageous to have the formative and reflective monitoring and evaluation to be part of a complex, participatory project from the outset. |

*Bold and italics = clear evaluation framework and/or criteria defined, with case study described*; **Bold = clear evaluation framework and/or criteria defined, with no case study**; *italics = evaluation process defined, with case study described*; *Italics and underline = clear evaluation framework, criteria and/or process defined, with no case study*.

adaptable to other PM programs [24, 25, 59]. However, for some this came at the cost of over-simplifying the evaluation framework [25, 55, 56, 61]. The strengths and limitations of each individual study are presented in Table 5. Recurring themes were synthesized from across the papers utilizing Freebairn's adapted 6P communications framework for reporting PM programs [13]. These themes are presented in Table 6.

## Part II: Development of a comprehensive multi-scale evaluation framework

The strengths and limitations of each individual study presented in Table 5 as well as the recurring themes synthesized in Table 6 informed the development of a comprehensive multi-scale

**Table 6. Recurring themes synthesized via Freebairn's adapted six-dimensional (6P) communications framework that standardize the reporting of PM studies.**

| Six-dimensional (6P) reporting criteria | Synthesized evaluation themes from included studies |
|---|---|
| Purpose (e.g., why PM approaches should be evaluated) | To develop an evaluation framework and/or criteria for PM programs and/or tools. |
| | The aims differed across studies, but ranged from evaluating the: success of PM programs with consideration of participatory processes [24, 25, 35, 58, 61]; changes in systems behavior to address complex challenges [60]; changes in stakeholder behaviors such as learning, decision making and values [57, 59]; PM tools [54, 55], and; difference between modelling outcomes and outputs [56]. |
| Process (e.g., method utilized to execute evaluation framework/criteria/process) | Not described in all studies. Various* evaluation methods described including: questionnaires [24, 55, 57, 61]; *ex post* evaluation [24, 25, 57]; *ex ante* evaluation [24, 57]; process evaluation [25, 55, 58]; evaluation forms [56]; workshop recordings [56, 57, 60]; semi-structured interviews [24, 35, 55, 58, 60, 93]; researchers' theoretical assessment [55, 60]; social network analysis [60]; participant observation [24, 60]; focus groups [24]; pre and post testing [24]; document analysis [24]; informal conversational meetings [24]; formative evaluation [24, 61], and; outcome evaluation [56]. |
| Partnerships (e.g., stakeholders involved in the development of the evaluation framework/criteria/process) | Not described in all studies. Mention of discussion-based process amongst authors [25, 54], and; workshop process with stakeholders [24]. |
| Products (e.g., evaluation approach–e.g., theoretical, conceptual, implementation) | Theoretical evaluation framework and/or criteria only with no presented application case study [54]; theoretical evaluation framework and/or criteria presented with case study described [25, 35, 55, 58, 59], theoretical evaluation framework, criteria, and/or process with no presented application case study [24], and; evaluation framework described as a process with case study [56, 57, 60, 61]. |
| imPact (e.g., outcomes/strengths of the evaluation framework/criteria/process) | Various* outcomes identified including: flexible framework to be applied to other disciplines [24, 54, 59, 61]; consideration of participatory process as well as influence of actual technical model [24, 35, 55]; supports decision making [24, 25, 54]; evaluation considers the whole process that can be adapted for each unique study's context and purpose [24]; evaluation of interventions in complex settings [60]; prompts action and reflection in evaluation design to cater to dynamic nature of complex systems [60, 61]; evaluates the engagement, learning and/or integration of stakeholders in PM [55, 57, 59]; evaluation of PM outcomes and outputs [24, 35, 55, 56], and; consideration of short- and long term outcomes [24, 58]. |
| Prioritizing (e.g., barriers/future opportunities of the evaluation framework/criteria/process) | Various* barriers/future opportunities identified including: limited description regarding the empirical evidence utilized to develop the presented evaluation framework [54, 57]; evaluation framework, criteria and/or process arguably oversimplified [25, 55, 56, 61]; evaluation framework development and/or application not discussed in depth [35, 58, 60]; evaluation focuses solely on the participatory process without factoring how the participatory process influences the model implementation [59], and; no demonstrated effectiveness of the evaluation framework through case study [24, 60]. |

*Some studies may be mentioned more than once, as they have employed more than one approach.

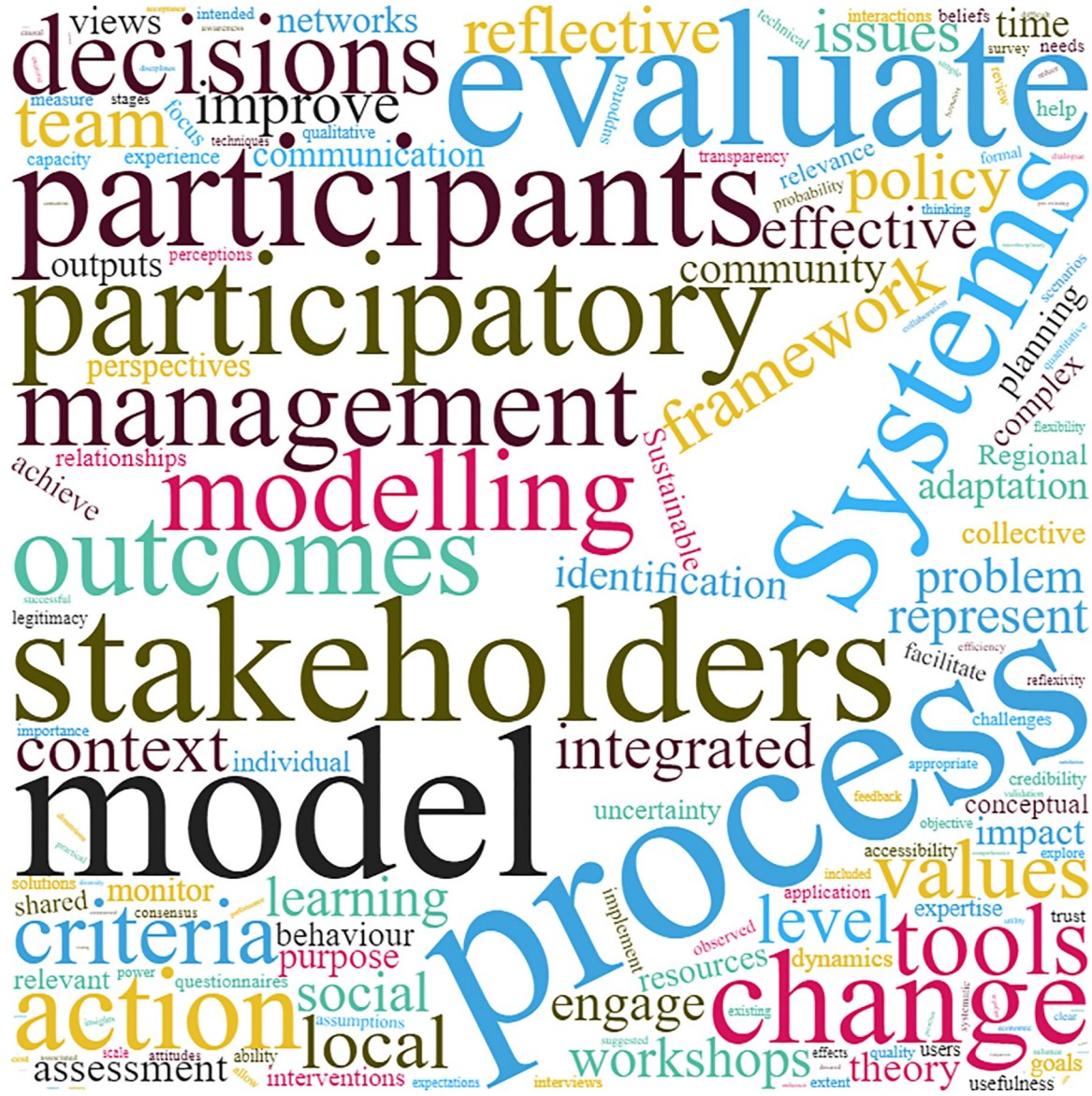

**Fig 2. Word cloud generated by uploading the full text of all studies included for synthesis.**

evaluation framework. This information was supplemented through the development of a word cloud, which allowed all authors to engage in an iterative process to refine and finalize the presented framework (Fig 2).

The synthesis process of the scoping review revealed opportunities to develop an evaluation framework that builds off empirical evidence of existing literature. For example, to develop a framework that can be flexibly adapted for diverse PM programs regardless of the discipline or modelling method [24, 54, 59, 61], considers the whole participatory process [24, 25, 35, 54, 55], and evaluates the engagement, learning as well as the integration of stakeholders in the PM process in both the short- and long-term [24, 35, 55–59]. A flexible framework allows for a mixed-methods approach to support real-world implementation [24, 25, 35, 55–58, 60, 93], which is described in further detail below in the *Discussion*.

It was evident from the synthesis process that differing terminologies, approaches, and assumptions were used, which led to challenges, including PM evaluation efforts being siloed from one another. This poses risk that evaluation processes will not reach their full potential, with associated implications for funders, participating stakeholders, as well as modellers [94]. Additionally, it was evident during the synthesis process that studies either described a comprehensive evaluation framework, criteria, and/or process, or they focused on the actual evaluation methodologies; the two rarely coincided [95, 96]. Therefore, a comprehensive evaluation framework and criteria are needed for PM programs that have theoretical and empirical underpinnings but are also accompanied by practical evaluation tools and methods.

A total of 40 unique evaluation terms were identified as the most utilized across the studies included for synthesis. *Process* appeared the most frequently (623 times), whereas *inclusive* appeared the least (six times). There are limitations to word clouds, for example, they arguably provide only a superficial snapshot of themes across studies. Therefore, all authors engaged in an iterative discussion process, utilizing the word cloud presented in Fig 2 as a discussion tool to enable more in-depth exploration of the final list of terms to further analyze and incorporate into the presented PM evaluation framework. Four broad evaluation framework categories were identified, based on the key themes and limitations described above as part of the scoping review synthesis process–*feasibility*, *value*, *change & action (impact)* and *sustainability* (highlighted in yellow, Table 7). As an evaluation concept, the *feasibility* or plausibility of PM allows for questions to be asked regarding whether it was possible for all participants to engage and contribute throughout the PM process. Consideration of the *value* of the PM process allows for the exploration of questions regarding what was gained due to engaging participants in PM (e.g., learning, confidence, trust). *Change & action* facilitates observations of impact, including *ex ante* and *ex post* comparisons of stakeholder relationships, knowledge, and behaviors as a result of the PM process; *sustainability* allows for the observations of PM outcomes over time (Fig 3). For clarity, it is acknowledged that PM is conducted in a dynamic (changing) environment, and sustainability of impacts may not always be desired. Thus, by *sustainability*, we refer to the observation of longer-term (prolonged) outcomes of the *feasibility*, *value*, and *impacts*, and not necessarily that these outcomes must remain stagnant.

The remaining 35 terms (highlighted in grey, Table 7) identified during the development of the word cloud have been incorporated as evaluation criteria, presented as evaluation questions in Fig 3. The word *level* was neither incorporated as an evaluation framework category nor criteria; but as the various levels of evaluation (e.g., multi-value perspectives) were noteworthy across studies, this term was included as a separate component in our evaluation framework–specifically, consideration of project-, individual-, group-, and system-level impacts (Fig 3).

As recognized by Jones et al. [25], Hamilton et al. [24], and Zorilla et al. [55], the consideration of multiple evaluation perspectives, or multiple levels of impact from the project-, individual-, group- to system-level is important as PM processes are increasingly becoming more inclusive to involve stakeholders from diverse backgrounds (e.g., client vs decision makers). Our framework further explores consideration of multiple evaluation perspectives, by recognizing that sublevels can exist within the individual- and group-levels (Fig 3). Consideration of the sublevels of participation enables the recognition of, for example, potential power relations and dynamics amongst the stakeholders, to be able to improve PM design and appropriately measure outcomes [24, 25, 54–56, 59]. This has been reflected in our evaluation framework, presented in Fig 3, with the individual- and group-levels further stratified to include community participants (e.g., consumer representatives) and professional participants (e.g., policy makers). As the differentiation of community and professional stakeholders is not always

**Table 7. Words extracted for development of a comprehensive multi-scale evaluation framework.**

| Term | # of times the term appeared across studies |
|---|---|
| process | 623 |
| decision / decision making | 461 |
| change | 333 |
| outcome | 304 |
| action | 301 |
| value | 212 |
| context | 181 |
| level | 175 |
| reflection | 170 |
| policy | 164 |
| effective | 157 |
| learn | 141 |
| engage | 138 |
| time | 106 |
| impact | 102 |
| network | 98 |
| communication | 95 |
| planning | 92 |
| resource | 89 |
| behavior | 89 |
| sustainable | 82 |
| feasible | 78 |
| relationship | 67 |
| belief | 52 |
| trust | 52 |
| capacity | 51 |
| needs | 51 |
| accessibility | 50 |
| credibility | 50 |
| transparency | 47 |
| feedback | 36 |
| flexibility | 36 |
| efficiency | 32 |
| simple | 28 |
| validation | 28 |
| acceptance | 26 |
| salience | 26 |
| utility | 26 |
| accuracy | 8 |
| inclusive | 6 |

Yellow highlight suggests common themes that have formed the overall evaluation framework categories; grey highlight suggests common sub-themes that have been incorporated as evaluation criteria; green highlight suggests the consideration of multiple evaluation levels (e.g., project vs system level) to be embedded within the evaluation framework.

| | FEASIBILITY *Is PM feasible?* | VALUE *What is the value of the PM process?* | CHANGE & ACTION (IMPACT) *What changed as a result of PM?* | SUSTAINABILITY *What are the outcomes of PM over the longer-term?* |
|---|---|---|---|---|
| **PROJECT-LEVEL IMPACT** | Is it feasible to develop systems models through participatory methods? *(Criterion 1)* <br> Is it feasible to recruit all necessary stakeholder perspectives in the PM process? *(Criterion 2)* | How did the PM process add value (e.g., context, validity, learning, salience) to developing the systems models? *(Criterion 10)* <br> What are facilitators and barriers to develop systems models through participatory methods (e.g., incentives, time, resources)? *(Criterion 11)* | How was feedback considered throughout the program to improve the PM process (including the build of the systems model)? *(Criterion 17)* <br> Was the PM process flexible enough to take action/respond to the changing needs of the complex system? *(Criterion 18)* | How does the PM process promote sustained use of the systems model? *(Criterion 25)* |
| **INDIVIDUAL-LEVEL IMPACT** | | | | |
| **Community Participants (e.g., consumer representatives)** | How do community participants view the credibility of the PM process? *(Criterion 3)* <br> How did the community participants contribute and engage during the PM process? *(Criterion 4)* | What are the experiences (e.g., benefits and challenges) arising from the application of PM to community participants (e.g., positive outcomes, ability to share their story)? *(Criterion 12)* | Are there changes in/what are the impacts in perceived knowledge, beliefs, behaviors, or assumptions for participants? *(Criterion 19)* <br> Are there changes/what are the impacts in the way participants engage with the system (e.g., reflection)? *(Criterion 20)* | Are there sustained changes in knowledge, beliefs, behaviors and/or assumptions for participants (e.g., resilience to uncertainty)? *(Criterion 26)* |
| **Professional Participants (e.g., policy makers)** | How do professional participants view the credibility of the PM process? *(Criterion 5)* <br> How do professional participants view the credibility of the evidence used to effectively inform the systems model? *(Criterion 6)* | What are the experiences (e.g., benefits and challenges) of professional participants using the systems model (e.g., confidence using the tool, ease/simple to use, acceptance)? *(Criterion 13)* | | |
| **GROUP-LEVEL IMPACT** | | | | |
| **Community Participants (e.g., consumer groups)** | How were power relationships managed? *(Criterion 7)* | What are the experiences (e.g., benefits and challenges) working in collaboration with professional participants during PM (e.g., communication, relationships, trust, social networks)? *(Criterion 14)* | Are there changes in/what are the impacts of social network connections and interdisciplinary collaboration as a result of the PM process? *(Criterion 21)* | Are changes in social network connections and interdisciplinary collaborations sustained over time? *(Criterion 27)* |
| **Professional Participants (e.g., policy makers)** | How did professional participants ensure that PM processes were inclusive, accessible, and transparent? *(Criterion 8)* | What are the experiences (e.g., benefits and challenges) working in interdisciplinary collaboration with community participants and/or other professional participants during the PM process (e.g., communication, relationships, trust, social networks)? *(Criterion 15)* | Are there changes/what are the impacts in perceived knowledge, beliefs, behaviors or assumptions for broader stakeholders (e.g., organizational learning)? *(Criterion 22)* | |
| **SYSTEM (POLICY)-LEVEL IMPACT** | Can systems models be built through a participatory approach that can effectively inform policy, planning, and investment decisions with a degree of confidence in accuracy to address complex systems challenges? *(Criterion 9)* | Does the participatory approach in building systems models add sufficient value to warrant the time and resources investment (e.g., improve capacity/efficiency, confidence)? *(Criterion 16)* | How have insights from the PM process been applied in the complex system of interest? *(Criterion 23)* <br> What are the factors that have influenced the extent to which the systems model has been utilized? *(Criterion 24)* | How have insights from the systems models been applied in the longer term? *(Criterion 28)* <br> How do participants' engagement with and use of the systems model change over time? *(Criterion 29)* <br> What are the longer-term factors that have influenced the extent to which the systems model is ongoingly utilized to inform policy, planning, and investment decisions? *(Criterion 30)* |

**Fig 3. Comprehensive multi-scale evaluation framework for PM programs.**

possible, the evaluation framework and criteria has been developed to be adapted for different PM program contexts.

## Discussion

This scoping review identified 11 studies that described an evaluation framework, criteria and/or process developed for PM programs. From the synthesis of these papers, the strengths and limitations, as well as overlapping concepts and themes were synthesized and analyzed to inform the development of a comprehensive multi-scale evaluation framework (Fig 3) that is designed to be adaptive, flexible, and iterative for PM programs, regardless of the discipline of study or modelling method. Such a framework is desirable as PM evaluation practices are currently limited across all fields. Our framework consists of four categories–(i) *Feasibility;* (ii) *Value;* (iii) *Change & Action (Impact)*, and; (iv) *Sustainability*. It is recommended that comprehensive evaluation processes need clear criteria to set appropriate benchmarks [55, 73]; therefore, the authors developed 30 criteria–presented as questions–which also include all key words identified from the word cloud (Fig 3). Though the word cloud was useful to identify commonly used terminology and themes, word clouds are limited in only providing a superficial overview. This prompted all authors to engage in an iterative process over three months to refine and finalize the presented evaluation framework.

There is recognition that many evaluation practices have been inadequate in both depth and scope which has limited the ability to improve PM practices [24, 31, 35]. The development of the presented evaluation framework through the synthesis of the 11 studies on PM evaluation provides the opportunity to draw on the expertise from other authors–ensuring the

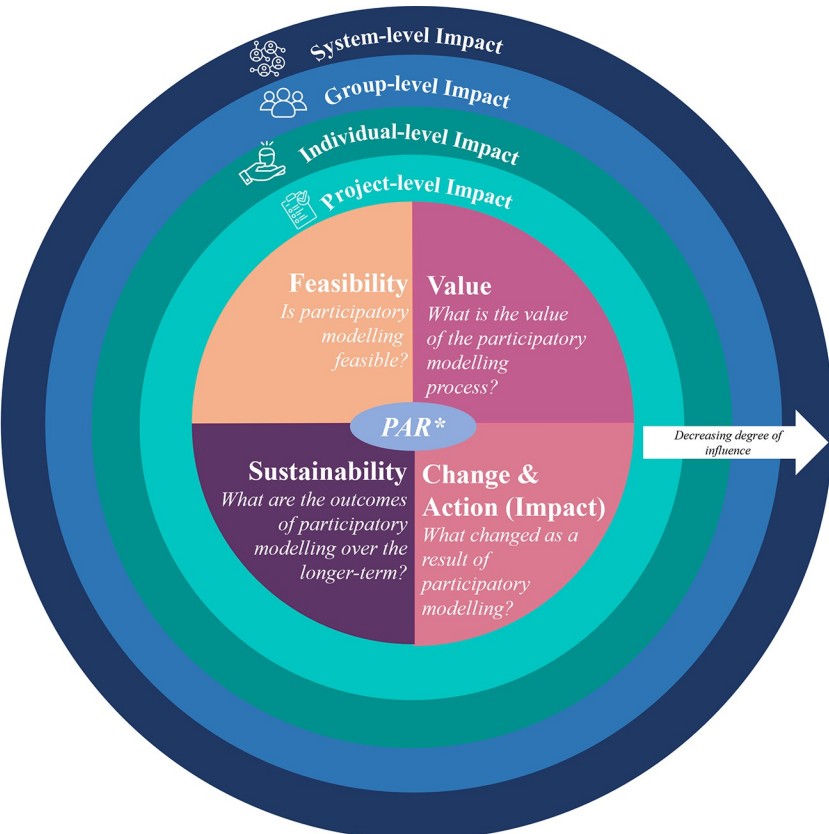

*PARTICIPATORY ACTION RESEARCH (PAR): Culture of embedding reflection & action during evaluation processes to improve outcomes and reduce inequities*

**Fig 4. Evaluation framework underpinned by participatory action research, with consideration of impacts across the project-, individual-, group-, and system-level.**

presented framework is guided by the identified strengths, challenges, and opportunities of existing work.

The presented evaluation framework was also developed with the consideration of the various aims in PM evaluation, as identified in the 11 studies included for synthesis. Specifically, our evaluation framework explicitly includes criteria that enables the observation of changes in stakeholder behaviors such as learning and decision making (*Criteria 9–10, 19–22*) [57, 59]; evaluation of the success of PM in the context of the participatory process (*Criteria 1, 7–8, 16*) [24, 25, 35, 58, 61]; assessment of the changes in systems behavior to address complex challenges (*Criteria 9, 16, 23–24, 28–30*) [60]; evaluation of PM tools (*Criteria 3–6*) [54, 55], and; the consideration of differences between modelling outcomes and outputs (*Criteria 23, 28–29*) [56]. Our evaluation framework builds on the empirical work of others, taking into consideration the participatory process as well as influence of actual technical model [24, 35, 55]; providing a flexible framework that can be applied to other disciplines [24, 54, 59, 61]; enabling the observation of short- and long term outcomes [24, 58], and; prompting action and reflection in evaluation design to cater to dynamic nature of complex systems [60, 61].

Enabling action and reflection is of particular importance to ensure improvement throughout the PM process. As such, our proposed evaluation framework (which include all 30 evaluation criteria) is underpinned by the principles of Participatory Action Research (PAR) (Fig 4). PAR embeds reflection during all phases of the PM program and can lead to shared learning and joint action for change to improve PM processes. PAR is a bottom-up approach and is

appropriate in the context of PM as the traditional roles of the modellers as the experts and stakeholders as the study participants are challenged [97, 98]. By working with the people who the modelling most affects (such as consumer representatives), PM outcomes can be improved through a more equitable process [99].

## Application of the presented evaluation framework

The studies included for synthesis (Tables 5 and 6) used a variety of methods to collect evaluation data. It is recommended that the presented evaluation framework adopts a mixed methods approach to align with the PM process. Examples of the potential methods include semistructured interviews, surveys, journey maps and social network analysis (Fig 5).

As the presented evaluation framework was developed to have broader international applications across disciplines and diverse participatory modelling programs, a more thorough

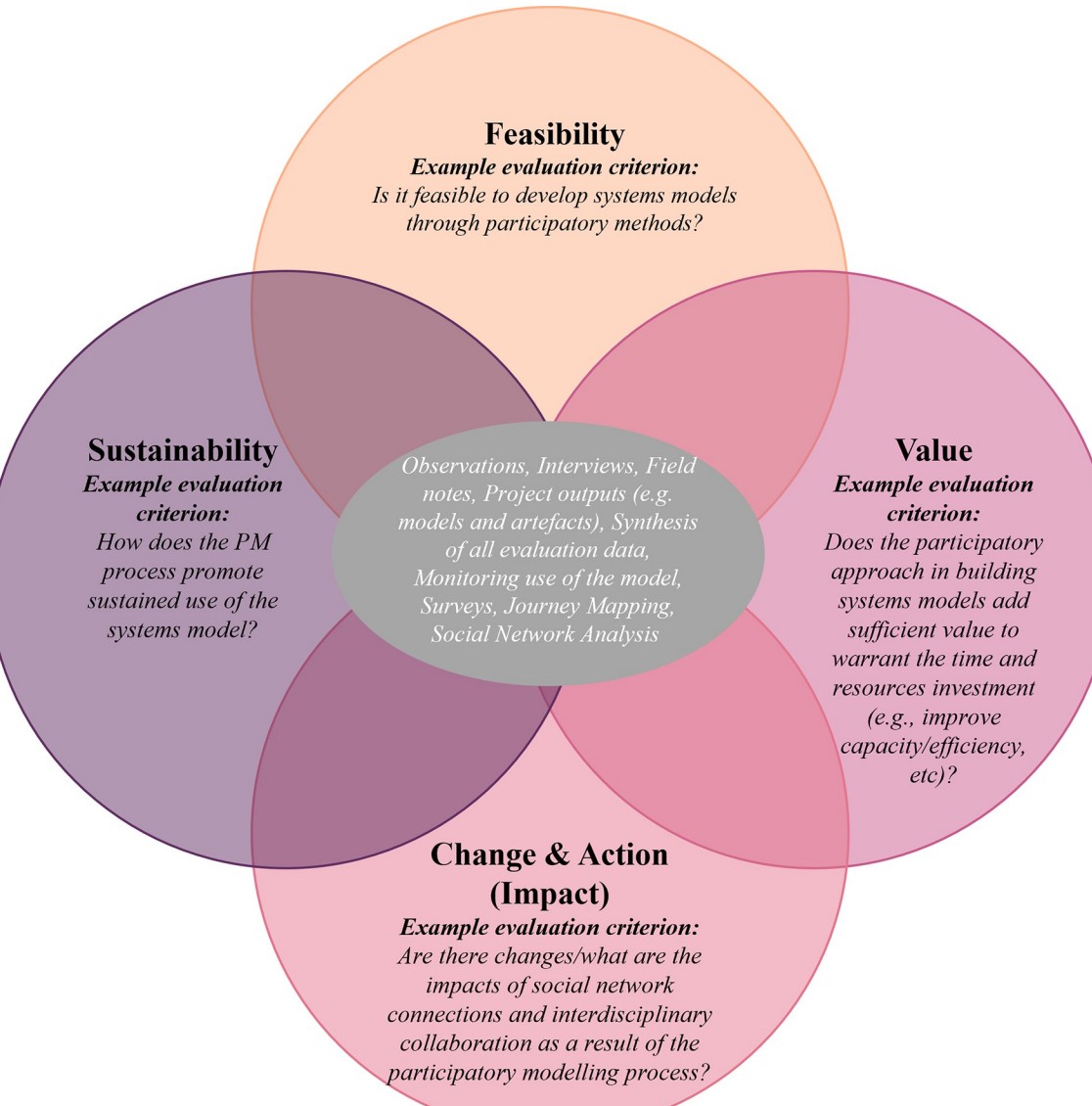

**Fig 5. Potential methods of data collection to adopt the comprehensive multi-scale framework into evaluation practice.**

description of how the authors plan to deploy the framework through a mixed-methods approach is described elsewhere. This description includes the tools developed as well as the suggested evaluation time points (*ex ante* and *ex post*) in the context of national multi-site youth mental health participatory systems modelling program (*Right care*, *first time*, *where you live* research Program). Further details of the research Program, including our participatory modelling approach, are also described elsewhere [100–102].

In summary, this PM program will develop system dynamics models for youth mental health across eight diverse regions across Australia. The evaluation framework has been translated into online survey and semi-structured interview questions, and will be underpinned by PAR principles, as well as formative, summative, process, and impact evaluation techniques. Novel research methods such as the gamification of online surveys, will enable unique data analysis (e.g., social network analysis), supporting the exploration of diverse stakeholder experiences to improve the PM process.

## Strengths, limitations and opportunities for future research

Evaluations have the potential to measure change at the project, individual, group, and system (policy) level to improve understanding of the PM process and the elements that facilitate certain outcomes (e.g., support decision making) [24, 103, 104]. In other words, knowledge that is acquired from evaluation outcomes can be applied prospectively throughout the PM process, rather than retrospectively reflecting on before-after measurements. Through principles of PAR, the proposed evaluation framework embeds continuous cycles of reflection to facilitate shared learning and iterative refinement of processes throughout PM implementation. Careful thought on design aspects are needed to ensure that evaluations are worthwhile as they require additional time, resources, and funding [105]. The presented evaluation framework considers the contributions of all participants involved in the PM process, not only the perspectives of the modellers or funder [105, 106].

The presented evaluation framework is also designed to be adaptive, flexible and iterative, to ensure that the framework remains relevant in the evolving fields and contexts in which PM are being applied [9]. PM by nature is a subjective process that is largely dependent on social interactions, human beliefs, biases and values. To ensure rigor throughout the PM evaluation process, the proposed evaluation framework builds on principles of PAR to empower stakeholders from various backgrounds (e.g., community participants to professional participants), and embeds ongoing reflection and learning so that the PM process can respond to the changing needs of complex systems to ensure that the aims of PM are met–on collaboration, learning, communication, as well as be applied across disciplines and diverse modelling methods [21]. The presented evaluation framework supports the application of a mixed-methods approach with an emphasis on approaching PM evaluations holistically [107].

There are limitations to this scoping review that should be acknowledged. The heterogeneity in terminology was a challenge during the screening, data extraction, and full text review process. Thus, with the process described in which the first author and senior author worked closely to resolve any ambiguity, a robust method was followed to ensure that the studies included are most relevant for the purposes of this scoping review. Additionally, though the described search strategy was broad in that it did not set any limits to the field of study, it was narrow so that only the studies that disclosed an evaluation framework, criteria and/or process in a PM context were included. The choice of database and search terms will also have its limitations. For instance, as our initial search was conducted in May 2021, it is possible that additional literature has been published since then that may meet the inclusion criteria for our scoping review. However, the presented evaluation framework provides a comprehensive

synthesis of and builds on the expertise of existing work, adding valuable contributions in the field of PM evaluation.

Additionally, the authors acknowledge that implementation of this framework in different contexts may mean that some aspects of it may be emphasized while other aspects de-emphasized. Our framework provides a uniquely comprehensive lens and a necessary contribution to the PM evaluation literature as an attempt to encourage researchers to consider evaluations in PM programs as a standard process of scientific inquiry.

## Conclusions

Evaluations are an integral component of the PM process that should be carefully considered throughout, and not viewed as its own separate component or afterthought. With the ability to inform policy change by demonstrating the measured effectiveness of PM, such processes should be adequately supported with an appropriate evaluation design. The presented framework describes a multi-scale and comprehensive, yet flexible evaluation approach that is built on the rigorous synthesis of strengths and opportunities for further development identified from existing studies. This framework enables the conduct of holistic evaluation practices by considering the project-, individual-, group-, and system-level impacts to understand the feasibility, value, impact, and sustainability of the PM process. Outputs from adopting such an evaluation approach, underpinned by principles of PAR, can be used to guide ongoing improvements to the PM process, empower stakeholders and users of systems models to be more confident in the model outcomes, as well as to improve understanding of which aspects of PM are particularly important for policy decisions.

## Supporting information

**S1 File. PRISMA Scoping review checklist.**
(DOCX)

## Acknowledgments

The authors would like to thank Glen Smyth, Academic Liaison Librarian at The University of Sydney, for his guidance and support in developing the search strategy. The authors would also like to thank Chloe Wilson for her support in assisting with the initial literature search.

## Author Contributions

**Conceptualization:** Grace Yeeun Lee, Ian Bernard Hickie, Jo-An Occhipinti, Yun Ju Christine Song, Louise Freebairn.

**Formal analysis:** Grace Yeeun Lee.

**Funding acquisition:** Ian Bernard Hickie, Jo-An Occhipinti, Yun Ju Christine Song.

**Methodology:** Grace Yeeun Lee, Louise Freebairn.

**Supervision:** Louise Freebairn.

**Writing – original draft:** Grace Yeeun Lee.

**Writing – review & editing:** Ian Bernard Hickie, Jo-An Occhipinti, Yun Ju Christine Song, Adam Skinner, Salvador Camacho, Kenny Lawson, Adriane Martin Hilber, Louise Freebairn.

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
