## [Decision Letter · Decision Letter 0]

15 Nov 2021

PONE-D-21-25695Presenting a comprehensive multi-scale evaluation framework for participatory modelling programs: A scoping reviewPLOS ONE

Dear Dr. Lee,

Thank you for submitting your manuscript to PLOS ONE. After careful consideration, we feel that it has merit but does not fully meet PLOS ONE’s publication criteria as it currently stands. Therefore, we invite you to submit a revised version of the manuscript that addresses the points raised during the review process.

We look forward to receiving your revised manuscript.

Kind regards,

Fausto Cavallaro, PhD

Academic Editor

PLOS ONE

Journal Requirements:

This research is being conducted under the Brain and Mind Centre’s Right care, first time, where you live Program, enabled by a AUD12.8 million partnership with BHP Foundation. The Program will develop infrastructure to support decisions relating to advanced mental health, and guide investments and actions to foster the mental health and wellbeing of young people in their communities.

3. Thank you for stating the following in the Competing Interests/Financial Disclosure * (delete as necessary) section: 

Dr Louise Freebairn is currently employed part-time by the Brain and Mind Centre, University of Sydney; ACT Health as Director of Knowledge Translation and Health Outcomes, Epidemiology Section, ACT Health & CSART as Director of Policy Applications & Translational Science. 

A/Professor Jo-An Occhipinti is both Head of Systems Modelling, Simulation & Data Science at the University of Sydney's Brain and Mind Centre and Managing Director of Computer Simulation & Advanced Research Technologies (CSART).

Professor Ian Hickie is the Co-Director, Health and Policy at the Brain and Mind Centre (BMC) University of Sydney. The BMC operates an early-intervention youth services at Camperdown under contract to headspace. He is the Chief Scientific Advisor to, and a 5% equity shareholder in, InnoWell Pty Ltd. InnoWell was formed by the University of Sydney (45% equity) and PwC (Australia; 45% equity) to deliver the $30 M Australian Government-funded Project Synergy (2017-20; a three-year program for the transformation of mental health services) and to lead transformation of mental health services internationally through the use of innovative technologies.

We note that you received funding from a commercial source: Brain and Mind Centre, University of Sydney

Reviewers' comments:

Reviewer's Responses to Questions

**Comments to the Author**

1. Is the manuscript technically sound, and do the data support the conclusions?

Reviewer #1: Partly

Reviewer #2: Yes

Reviewer #3: Partly

Reviewer #4: Yes

2. Has the statistical analysis been performed appropriately and rigorously? 

Reviewer #1: Yes

Reviewer #2: Yes

Reviewer #3: N/A

Reviewer #4: Yes

3. Have the authors made all data underlying the findings in their manuscript fully available?

Reviewer #1: Yes

Reviewer #2: Yes

Reviewer #3: Yes

Reviewer #4: Yes

4. Is the manuscript presented in an intelligible fashion and written in standard English?

Reviewer #1: Yes

Reviewer #2: Yes

Reviewer #3: Yes

Reviewer #4: Yes

5. Review Comments to the Author

Reviewer #1: Evaluation of participatory modelling (PM) cases and processes is an impotent yet greatly neglected handled issue. Based on a recent review of PM case studies we know that a great number of studies does not present an evaluation at all, and that case studies that do, often do not report clear assessment criteria nor methods of data collection and analysis or the grounds for choosing them. To my view, evaluation should be a regular feature of scientific enquiry.

Therefore this paper is valuable. Mainly, it is also well written and structured and the idea and design is justified in relation to its objectives. There are however some issues that needs to be considered before publication:

Presentation: Good language and (mainly good) disposition. But: the background section (especially the one about evaluation) is not properly integrated – it is not used to explain the study’s approach to evaluation, and the discussion, of the framework does not make use of it. Moreover – the explanation of the method for making the synthesis needs to be moved to the method section and expanded. Finally, the discussion section includes parts describing the developed framework, which would better suit the result section. The discussion could also make much better use of literature on evaluation and of the literature underpinning the framework (see comment below about missing literature connection)

The understanding (perspective) of PM: Seems to be a focus here on the (participatory) model/modelling itself as compared to focusing on participatory planning and decision-making (including (participatory) modelling. Be clear about which perspective this study focuses on, because it may make a great difference in terms of what an evaluation can or should be targeting. (Modell building process or societal planning/decision-making process)

Search method (objective 1): Good search and selection method and also described in a very good way.

Understanding of the identified evaluation frameworks (objective 2): Method described well.

Seems that the authors have gained a good understanding of the frameworks as judged by the presentation

Synthesis of knowledge/frameworks to develop a new broad(er?) framework (objective 3): Method description too limited in the methods section. “Narrative synthesis”? Include reference, and most importantly a description of the procedure including the use of the 6P framework, the word-count etc and how they all contribute to the synthesis.

Furtermore, the authors claim that the new framework is based on a ”synthesis of expertiese”. As I understand it, this expertise is represented by the article words of the the 11 identified frameworks/articles. But - there are so much underlying knowledge that is represented by/in these frameworks. This knowledge cannot be captured by a word count/cloud of their presentations solely. Please discuss this problem and how you wish to handle it. I feel a connection to the vast theoretical and empirical literatures that underlie the suggested framework is missing. I understand this might be much to ask – you already do so much here. But linking to the underlying knowledges is important for the future usefulness of your framework. The user needs to understand more about the knowledge that underlies your questions. One possibility could be to divide the paper in to two papers. One with the review (objective 1 and 2) and one with the development of the new framework (objective 3) which then would have room for proper linkage to literature on participatory planning and decision-making processes, and evaluation.

One other, related, problem (which, if the authors agree, can be handled through re-formulations at some places) is the idea of finding/defining the “best” framework for PM evaluation. This idea shines through the text at some places. E.g. in line 327 the idea of is prevalent, but also in the title (comprehensive framework). Any evaluation framework will provide one particular understanding of PM, not “the one and only true” understanding of PM. Sometimes the perspective of a framework is clearly and explicitly described, but often, I find that underlying assumptions and perspectives are not. One may have the intention to cover more or less aspects of PM in an evaluation, but as the study implies (line 318), the ability to be broad generally comes at the cost of being less specific/detailed.

Connected to this, I really like the idea and formulation that describes the intention of developing a framework that can be adopted to different PM contexts and evaluation purposes. But still, what you include in your “comprehensive” framework should not be forwarded as the framework. Like the frameworks in your review it will have weaknesses and strengths. It would be good in you formulated there your selves.

Finally - I hope this is helpful for you. To aid the understanding of my comments above you may also look at the comments made in the manuscript which I have attached to this review.

Good luck.

Reviewer #2: Evaluation of various methods used in participatory research in general, and in participatory modeling (PM) in particular has always been challenging. The paper is certainly a useful step in the right direction and is worth being published. Please see attached manuscript with comments and suggestions.

My main concern is that the proposed framework remains quite theoretical and there is no attempt to show how it can work in practice. There is some mention of some applications that are published "elsewhere" with no clear reference to follow. As a result it remains unclear how the evaluation can really work. Is there any way it can be used to compare various PM projects? What are the metrics of success?

It might be useful to acknowledge that in PM there will always be a good deal of subjectivity, as in any processes largely dependent on social interactions, human beliefs, biases and values. All this makes it only more difficult to develop an objective evaluation mechanism. After all PM is goal driven and the only real evaluation is not about how well it was structured and organized, but about how well the process helped to make better decisions and solve the problem. This might be quite hard to evaluate, because, again, it will be quite subjective and can very much depend on who is evaluating and when. It might take some time for this to become evident.

Reviewer #3: This paper presents a review of 11 papers that present an evaluation framework on participatory modelling, and a synthesizing evaluation framework. The aim to come up with an integrated evaluation framework is in itself valuable. However, the approach taken to develop the framework in the paper is not clear/convincing.

My main concern is that the paper consists of two parts that are rather unconnected. The first part is the review of literature on evaluation frameworks for participatory modelling. The second part is the evaluation framework. The authors are not clear on how the literature review has precisely informed the development of the framework. There is too little elaboration of the theoretical argument underlying the framework. It seems as if the framework could have been designed without the review; or based on a more general/unstructured lit review that is typically part of an academic paper (i.e. describing the state-of-the art and explain what is missing). After having read the paper, I get the feeling that presenting the paper as a review is a bit of a misguidance; it reads more as a lit review that has been conducted in the preparation of the development of an evaluation framework that may have been very useful to the authors in formulating ideas but that may not be the most important thing to present in this paper as such. Since the authors rely mainly on the paper by Hamilton et al, basically this paper could have been taken, leave aside all the rest, and still formulate the same framework. The framework itself is not explained enough, as a result of which it is hard to assess its validity. To be honest, I don’t see the rigor that the authors want to be there; rather, definitions are lacking, argumentation is limited/short, and a theory of evaluation is missing.

My advice would be to rewrite the paper focusing on the evaluation framework. This would mean reducing the scoping review part, and use that to write a theoretical section in the paper, and expand the framework part of the paper; explaining more and better how you arrived at this, and what each cell in the framework means/how it was selected and how it is defined.

More detailed comments:

Scoping review: it does not become clear from the paper what a scoping review exactly is, and how it differs from a systematic review. Also, it does not become clear why a systematic review would not be more suitable here.

P4 line 99: I would say this is not restricted to ‘local’ expert knowledge

L139: “Applying this definition to the context of this paper, there is opportunity to better understand the quality, importance, and value of PMs.” � From the introduction it does not become clear if this is a true gap; what has been done already on this topic?

The third objective of the review is to develop a framework “regardless of discipline or modelling method”; it would be good to explain why such a generic approach is desirable and feasible

L194: “and Medical Subject Headings” � doesn’t this conflict with the objective to include all disciplines?

Table 1: why is “co-design” a search term? And not other terms like co-creation, co-production etc, with and without hyphen. In general; the search terms need more clarification/explanation.

228: please explain what a data extraction template is and what it is used for

230-240: this part is not very clear

Table 7: the legend explains the meaning of the marked text in the table. This needs more explanation; it is now to so clear.

379-380: “conducting PM to develop an evaluation framework that consists of four categories – (i) Feasibility; (ii) Value; (iii) Change & Action (Impact), and; (iv) Sustainability.” It is not clear how this follows from the results

Table 8, feasibility/project level: is this an ex ante evaluation question?

Table 8 Sustainablity/project level: is that the ultimate aim? Isn't the model a means towards something else? e.g decision support?

Table 8 feasibility/community participants: why and how do you make the distinction between community participants and professional participants? That is only meaningful in specific situations I guess

Table 8 “How did the community participants contribute and engage during the PM process” � why is this under group level impact and not under individual?

Table 8: system level impact how do you define that?

P433: the paper mentions here the context of mental health; this is mentioned at other points in the paper as well. It is not clear if/how this context plays a role in the paper; can’t you do without? Especially since you find that the papers in the review are all in the domain of environmental management and planning.

L446: the authors are not explicit about the theory behind evaluation (which they find important themselves when selecting papers for inclusion in the review); why these levels, why these items, why these questions? How does PAR inform the framework precisely?

Reviewer #4: The review makes an interesting contribution by providing an evaluation framework method that may be adapted to PM in different science fields. The fact that the proposed method is designed to be adaptive, flexible and iterative and that the contributions of all participants are considered, are two characteristics that may be highlighted. However, although is out of the scope of the presented work, an application case would be necessary to fully appreciate and to assess the proposed method.

6. PLOS authors have the option to publish the peer review history of their article (what does this mean?). If published, this will include your full peer review and any attached files.

Reviewer #1: **Yes: **Beatrice Hedelin

Reviewer #2: **Yes: **A.Voinov

Reviewer #3: No

Reviewer #4: No

---

## [Author Response · Author response to Decision Letter 0]

9 Jan 2022

Fausto Cavallaro, PhD

Academic Editor

PLOS ONE

23 December 2021

Dear Dr Fausto Cavallaro,

Re Manuscript – Presenting a comprehensive multi-scale evaluation framework for participatory modelling programs: A scoping review (PONE-D-21-25695)

Thank you for your email dated 17 November 2021 providing us with Reviewer’s comments on manuscript (PONE-D-21-25695). 

Please find attached our revised manuscript, in which we have addressed all of the comments provided by the Reviewers as outlined in our point-by-point rebuttal below. We have provided a revised manuscript with modifications shown in tracked changes, as well as an unmarked version of our manuscript. 

We trust that the following satisfactorily addresses the Reviewers’ comments and that our revised manuscript is now suitable for publication in PLOS ONE, which is the most appropriate journal to publish our paper as the paper will be of great interest to researchers and experts across multiple disciplines. 

Many thanks and kind regards, 

Grace Yeeun Lee

Journal Requirements:

Our response: Many thanks for your guidance. We have ensured that all PLOS ONE style templates have been adhered. 

This research is being conducted under the Brain and Mind Centre’s Right care, first time, where you live Program, enabled by a AUD12.8 million partnership with BHP Foundation. The Program will develop infrastructure to support decisions relating to advanced mental health, and guide investments and actions to foster the mental health and wellbeing of young people in their communities. The funders had no role in study design, data collection and analysis, decision to publish, or preparation of the manuscript. 

Our response: We have updated our financial disclosure statement (changes highlighted above) – thank you so much for changing the online submission form on our behalf. 

3. Thank you for stating the following in the Competing Interests/Financial Disclosure * (delete as necessary) section: 

Dr Louise Freebairn is currently employed part-time by the Brain and Mind Centre, University of Sydney; ACT Health as Director of Knowledge Translation and Health Outcomes, Epidemiology Section, ACT Health & CSART as Director of Policy Applications & Translational Science. 

A/Professor Jo-An Occhipinti is both Head of Systems Modelling, Simulation & Data Science at the University of Sydney's Brain and Mind Centre and Managing Director of Computer Simulation & Advanced Research Technologies (CSART).

Professor Ian Hickie is the Co-Director, Health and Policy at the Brain and Mind Centre (BMC) University of Sydney. The BMC operates an early-intervention youth services at Camperdown under contract to headspace. He is the Chief Scientific Advisor to, and a 5% equity shareholder in, InnoWell Pty Ltd. InnoWell was formed by the University of Sydney (45% equity) and PwC (Australia; 45% equity) to deliver the $30 M Australian Government-funded Project Synergy (2017-20; a three-year program for the transformation of mental health services) and to lead transformation of mental health services internationally through the use of innovative technologies.

We note that you received funding from a commercial source: Brain and Mind Centre, University of Sydney

Our response: The Competing Interests Statements that we have submitted is corrected. The Brain and Mind Centre at The University of Sydney is not a commercial source. It is a multidisciplinary research centre that is not separate to The University of Sydney (which is a public university). 

Our response: We have updated our Supporting Information files at the end of our manuscript, and have updated the in-text citations to match accordingly. Many thanks for your guidance. 

Reviewers’ comments:

Reviewer #1:

Evaluation of participatory modelling (PM) cases and processes is an impotent yet greatly neglected handled issue. Based on a recent review of PM case studies we know that a great number of studies does not present an evaluation at all, and that case studies that do, often do not report clear assessment criteria nor methods of data collection and analysis or the grounds for choosing them. To my view, evaluation should be a regular feature of scientific enquiry.

Therefore this paper is valuable. Mainly, it is also well written and structured and the idea and design is justified in relation to its objectives. There are however some issues that needs to be considered before publication:

Presentation: Good language and (mainly good) disposition. But: the background section (especially the one about evaluation) is not properly integrated – it is not used to explain the study’s approach to evaluation, and the discussion, of the framework does not make use of it. Moreover – the explanation of the method for making the synthesis needs to be moved to the method section and expanded. Finally, the discussion section includes parts describing the developed framework, which would better suit the result section. The discussion could also make much better use of literature on evaluation and of the literature underpinning the framework (see comment below about missing literature connection)

The understanding (perspective) of PM: Seems to be a focus here on the (participatory) model/modelling itself as compared to focusing on participatory planning and decision-making (including (participatory) modelling. Be clear about which perspective this study focuses on, because it may make a great difference in terms of what an evaluation can or should be targeting. (Modell building process or societal planning/decision-making process)

Search method (objective 1): Good search and selection method and also described in a very good way.

Understanding of the identified evaluation frameworks (objective 2): Method described well.

Seems that the authors have gained a good understanding of the frameworks as judged by the presentation

Synthesis of knowledge/frameworks to develop a new broad(er?) framework (objective 3): Method description too limited in the methods section. “Narrative synthesis”? Include reference, and most importantly a description of the procedure including the use of the 6P framework, the word-count etc and how they all contribute to the synthesis.

Furthermore, the authors claim that the new framework is based on a “synthesis of expertise”. As I understand it, this expertise is represented by the article words of the the 11 identified frameworks/articles. But - there are so much underlying knowledge that is represented by/in these frameworks. This knowledge cannot be captured by a word count/cloud of their presentations solely. Please discuss this problem and how you wish to handle it. I feel a connection to the vast theoretical and empirical literatures that underlie the suggested framework is missing. I understand this might be much to ask – you already do so much here. But linking to the underlying knowledges is important for the future usefulness of your framework. The user needs to understand more about the knowledge that underlies your questions. One possibility could be to divide the paper in to two papers. One with the review (objective 1 and 2) and one with the development of the new framework (objective 3) which then would have room for proper linkage to literature on participatory planning and decision-making processes, and evaluation.

One other, related, problem (which, if the authors agree, can be handled through re-formulations at some places) is the idea of finding/defining the “best” framework for PM evaluation. This idea shines through the text at some places. E.g. in line 327 the idea of is prevalent, but also in the title (comprehensive framework). Any evaluation framework will provide one particular understanding of PM, not “the one and only true” understanding of PM. Sometimes the perspective of a framework is clearly and explicitly described, but often, I find that underlying assumptions and perspectives are not. One may have the intention to cover more or less aspects of PM in an evaluation, but as the study implies (line 318), the ability to be broad generally comes at the cost of being less specific/detailed.

Connected to this, I really like the idea and formulation that describes the intention of developing a framework that can be adopted to different PM contexts and evaluation purposes. But still, what you include in your “comprehensive” framework should not be forwarded as the framework. Like the frameworks in your review it will have weaknesses and strengths. It would be good in you formulated there your selves.

Finally - I hope this is helpful for you. To aid the understanding of my comments above you may also look at the comments made in the manuscript which I have attached to this review.

Good luck.

Our response: Thank you for your detailed review. To best address Reviewer #1’s feedback, we have: 

Presentation: 

• Rewrote our ‘Introduction’ section – specifically, on PM evaluation – to focus on our study’s approach to evaluation; 

• Expanded the ‘Materials and Methods’ section to elaborate on how we synthesized the literature to develop our evaluation framework; 

• Restructured our entire paper to provide a better flow of our ideas, addressing each of Reviewer #1’s suggestions, and; 

• Expanded the ‘Discussion’ section which now includes more reference to literature on PM evaluation, as well as the literature underpinning our framework.

The understanding (perspective) of PM:

• Explicitly declared the perspective of PM we are taking in our paper – aligned with our approach to developing a broad, adaptable and flexible evaluation framework, we propose to adopt a holistic consideration of the whole PM process, which includes knowledge integration and learning, technical systems model development, as well as participatory and integrated planning.

Synthesis of knowledge/frameworks to develop a new broad(er?) framework:

• Expanded the ‘Material and Methods’ section to provide more detail on the process for developing our evaluation framework – which included the use of the 6P framework and a word cloud (as well as disclosing limitations to our approach, and how limitations were addressed), and;

• Provided an explanation and reference of ‘Narrative synthesis.’

To address the reviewer’s suggestion to better connect the theoretical and empirical literature that underlie our framework, we have rewritten our ‘Discussion’ section. We have also expanded on the limitations of our framework, disclosing that though we aim to present a comprehensive evaluation framework developed through the synthesis of empirical knowledge, our framework is not the true/best framework, and will have its strengths and weaknesses. 

Reviewer #2:

Evaluation of various methods used in participatory research in general, and in participatory modeling (PM) in particular has always been challenging. The paper is certainly a useful step in the right direction and is worth being published. Please see attached manuscript with comments and suggestions.

My main concern is that the proposed framework remains quite theoretical and there is no attempt to show how it can work in practice. There is some mention of some applications that are published "elsewhere" with no clear reference to follow. As a result it remains unclear how the evaluation can really work. Is there any way it can be used to compare various PM projects? What are the metrics of success?

It might be useful to acknowledge that in PM there will always be a good deal of subjectivity, as in any processes largely dependent on social interactions, human beliefs, biases and values. All this makes it only more difficult to develop an objective evaluation mechanism. After all PM is goal driven and the only real evaluation is not about how well it was structured and organized, but about how well the process helped to make better decisions and solve the problem. This might be quite hard to evaluate, because, again, it will be quite subjective and can very much depend on who is evaluating and when. It might take some time for this to become evident.

Our response: 

We have reviewed Reviewer #2’s comments and suggestions and have addressed this in our manuscript.

We agree that this manuscript is theoretical. To demonstrate the application of the proposed evaluation framework, we have submitted a separate manuscript to PLOS ONE that is currently under review (PONE-D-21-25700), which details an evaluation study protocol to demonstrate how our evaluation framework can be applied in practice (in the context of a youth mental health participatory systems modelling program). We propose to publish the two manuscripts at the same time. 

We have expanded our ‘Discussion’ section and have disclosed that participatory modelling, by nature, is a subjective process that is largely dependent on social interactions, human beliefs, biases and values. However, we note the aims of participatory modelling – such as collaboration, learning, and communication – and how our evaluation framework has attempted to present an evaluation framework that comprehensively measures the aims of participatory modelling. 

Reviewer #3:

This paper presents a review of 11 papers that present an evaluation framework on participatory modelling, and a synthesizing evaluation framework. The aim to come up with an integrated evaluation framework is in itself valuable. However, the approach taken to develop the framework in the paper is not clear/convincing.

My main concern is that the paper consists of two parts that are rather unconnected. The first part is the review of literature on evaluation frameworks for participatory modelling. The second part is the evaluation framework. The authors are not clear on how the literature review has precisely informed the development of the framework. There is too little elaboration of the theoretical argument underlying the framework. It seems as if the framework could have been designed without the review; or based on a more general/unstructured lit review that is typically part of an academic paper (i.e. describing the state-of-the art and explain what is missing). After having read the paper, I get the feeling that presenting the paper as a review is a bit of a misguidance; it reads more as a lit review that has been conducted in the preparation of the development of an evaluation framework that may have been very useful to the authors in formulating ideas but that may not be the most important thing to present in this paper as such. Since the authors rely mainly on the paper by Hamilton et al, basically this paper could have been taken, leave aside all the rest, and still formulate the same framework. The framework itself is not explained enough, as a result of which it is hard to assess its validity. To be honest, I don’t see the rigor that the authors want to be there; rather, definitions are lacking, argumentation is limited/short, and a theory of evaluation is missing.

My advice would be to rewrite the paper focusing on the evaluation framework. This would mean reducing the scoping review part, and use that to write a theoretical section in the paper, and expand the framework part of the paper; explaining more and better how you arrived at this, and what each cell in the framework means/how it was selected and how it is defined.

More detailed comments:

Scoping review: it does not become clear from the paper what a scoping review exactly is, and how it differs from a systematic review. Also, it does not become clear why a systematic review would not be more suitable here.

P4 line 99: I would say this is not restricted to ‘local’ expert knowledge

L139: “Applying this definition to the context of this paper, there is opportunity to better understand the quality, importance, and value of PMs.” � From the introduction it does not become clear if this is a true gap; what has been done already on this topic?

The third objective of the review is to develop a framework “regardless of discipline or modelling method”; it would be good to explain why such a generic approach is desirable and feasible

L194: “and Medical Subject Headings” � doesn’t this conflict with the objective to include all disciplines?

Table 1: why is “co-design” a search term? And not other terms like co-creation, co-production etc, with and without hyphen. In general; the search terms need more clarification/explanation.

228: please explain what a data extraction template is and what it is used for

230-240: this part is not very clear

Table 7: the legend explains the meaning of the marked text in the table. This needs more explanation; it is now to so clear.

379-380: “conducting PM to develop an evaluation framework that consists of four categories – (i) Feasibility; (ii) Value; (iii) Change & Action (Impact), and; (iv) Sustainability.” It is not clear how this follows from the results

Table 8, feasibility/project level: is this an ex ante evaluation question?

Table 8 Sustainability/project level: is that the ultimate aim? Isn't the model a means towards something else? e.g decision support?

Table 8 feasibility/community participants: why and how do you make the distinction between community participants and professional participants? That is only meaningful in specific situations I guess

Table 8 “How did the community participants contribute and engage during the PM process” � why is this under group level impact and not under individual?

Table 8: system level impact how do you define that?

P433: the paper mentions here the context of mental health; this is mentioned at other points in the paper as well. It is not clear if/how this context plays a role in the paper; can’t you do without? Especially since you find that the papers in the review are all in the domain of environmental management and planning.

L446: the authors are not explicit about the theory behind evaluation (which they find important themselves when selecting papers for inclusion in the review); why these levels, why these items, why these questions? How does PAR inform the framework precisely?

Our response: 

Thank you for your thorough review. To best address Reviewer #3’s feedback, we have:

• Expanded our ‘Materials and Methods’ section to describe in more detail the process of how our evaluation framework was developed;

• Expanded the ‘Discussion’ section to better describe the theoretical arguments underlying the framework (which was enabled by the synthesis of literature via the scoping review); 

• Restructured our entire paper to better connect the scoping review with the presentation of our evaluation framework, and; 

• Addressed all detailed comments that Reviewer #3 provided.

On the reviewer’s suggestion that presenting the paper as a review is misguiding – we have elaborated a scoping review (which is defined by an a priori protocol) is most appropriate. We have also rewritten our ‘Discussion’ section to not rely so heavily on only Hamilton’s evaluation framework, but how all 11 studies included for synthesis in our scoping review has led to our current framework. 

Reviewer #4:

The review makes an interesting contribution by providing an evaluation framework method that may be adapted to PM in different science fields. The fact that the proposed method is designed to be adaptive, flexible and iterative and that the contributions of all participants are considered, are two characteristics that may be highlighted. However, although is out of the scope of the presented work, an application case would be necessary to fully appreciate and to assess the proposed method.

Our response: 

We have further emphasized that our proposed evaluation framework enables the consideration of contributions from diverse stakeholders through an adaptive, flexible and iterative approach. 

We also agree with Reviewer #4’s comment that an application case would be necessary to appreciate how to implement our evaluation framework. We have submitted a separate manuscript to PLOS ONE that is currently under review (PONE-D-21-25700) that details an evaluation study protocol to demonstrate how our evaluation framework can be applied in a PM program (in the context of a youth mental health participatory systems modelling program). We propose to publish the two manuscripts at the same time.

---

## [Decision Letter · Decision Letter 1]

18 Feb 2022

PONE-D-21-25695R1Presenting a comprehensive multi-scale evaluation framework for participatory modelling programs: A scoping reviewPLOS ONE

Dear Dr. Lee,

Thank you for submitting your manuscript to PLOS ONE. After careful consideration, we feel that it has merit but does not fully meet PLOS ONE’s publication criteria as it currently stands. Therefore, we invite you to submit a revised version of the manuscript that addresses the points raised during the review process.

We look forward to receiving your revised manuscript.

Kind regards,

Fausto Cavallaro, PhD

Academic Editor

PLOS ONE

Journal Requirements:

Reviewers' comments:

Reviewer's Responses to Questions

**Comments to the Author**

1. If the authors have adequately addressed your comments raised in a previous round of review and you feel that this manuscript is now acceptable for publication, you may indicate that here to bypass the “Comments to the Author” section, enter your conflict of interest statement in the “Confidential to Editor” section, and submit your "Accept" recommendation.

Reviewer #2: All comments have been addressed

Reviewer #3: (No Response)

2. Is the manuscript technically sound, and do the data support the conclusions?

Reviewer #2: Yes

Reviewer #3: Yes

3. Has the statistical analysis been performed appropriately and rigorously? 

Reviewer #2: Yes

Reviewer #3: N/A

4. Have the authors made all data underlying the findings in their manuscript fully available?

Reviewer #2: Yes

Reviewer #3: Yes

5. Is the manuscript presented in an intelligible fashion and written in standard English?

Reviewer #2: Yes

Reviewer #3: Yes

6. Review Comments to the Author

Reviewer #2: Good job reviewing the paper.

My only concern is about the second paper to be published simultaneously with the application example.

I am concerned in this case that there is no guarantee that both manuscripts will be accepted and published at the same time. Many journals would discourage this kind of two-part submission, but I would leave this to the PLOS editors to decide. It would be certainly nice if the paper would be self-sufficient.

A few minor notes:

- On the title page for three co-authors it is said that: ¶ These authors contributed equally to this work

Sounds a bit strange. And how did the other authors contribute?

- First sentence of the Abstract says: "Systems modelling and simulation can... better manage system challenges". Not sure what did you mean to say. How do you manage system challenges with modeling?

Reviewer #3: Dear authors,

Thank you for carefully addressing my comments. I think the paper has greatly improved.

I just have a few minor issues.

the evaluation framework that you have developed includes a time dimension; some of the criteria refer to ex ante assessments (e.g. criterion 2), some to ex durante (e.g.criterion 3) and others ex post (e.g. e.g. criterion 25). I think it would improve the evaluation framework if this time dimension is included explicitly, so basically showing what questions need to be asked at what moment in time?

Not all criteria are clear. E.g. criterion 11: What are facilitators and barriers (e.g. inemtives)? facilitators and barriers for what?

p4-193 this sentence is unclear :" Stakeholders who have a ‘stake’ in the system, including those who are a part of the local context are viewed as important communication agents to decision makers, facilitating a positive movement whereby stakeholder involvement is perceived as almost necessary – shifting scientists away from working independently to develop systems models [21]."

p8-180: systematic reviews also use a protocol...?

p16-339: this sentence is unclear " There were two papers that did not explicitly consider the multi-value perspectives integrated within PM [54, 56].

p36-514: "In other words, knowledge that is acquired from evaluation outcomes can be applied prospectively throughout the PM process, rather than retrospectively reflecting on before-after measurements." Can you say a bit more about how your framework can be used pro-actively, for the design of PM processes? That would be useful!

7. PLOS authors have the option to publish the peer review history of their article (what does this mean?). If published, this will include your full peer review and any attached files.

Reviewer #2: **Yes: **Alexey Voinov

Reviewer #3: No

---

## [Author Response · Author response to Decision Letter 1]

22 Feb 2022

Journal Requirements:

Our response: Many thanks for your guidance. We have reviewed the reference list to ensure that it is complete and correct. We have made the following changes:

- Corrected the author names for References #11, #29, #32. #38, #39, #40, #46, #50, #69, #71, #96. 

- Corrected the punctuations for References #39, #96. 

- Updated the registration details of the scoping review for Reference #49.

- Added details of the publishing city for Reference #69.

- Corrected the title for Reference #77. 

- Additionally added three new references #100-102.

Reviewers’ comments:

Reviewer #2:

Good job reviewing the paper.

My only concern is about the second paper to be published simultaneously with the application example.

I am concerned in this case that there is no guarantee that both manuscripts will be accepted and published at the same time. Many journals would discourage this kind of two-part submission, but I would leave this to the PLOS editors to decide. It would be certainly nice if the paper would be self-sufficient.

A few minor notes:

1. On the title page for three co-authors it is said that: “These authors contributed equally to this work.” Sounds a bit strange. And how did the other authors contribute?

2. First sentence of the Abstract says: "Systems modelling and simulation can... better manage system challenges". Not sure what did you mean to say. How do you manage system challenges with modeling?

Our response: Thank you for your kind feedback. We agree that the two papers would be best presented together, and thus have requested PLOS ONE to publish simultaneously if this will not substantially delay the publication of this manuscript. Regardless of the journal’s decision, we will ensure that the two papers appropriately reference each other. We have left a note for the PLOS ONE Academic Editor for further consideration. 

To address your additional notes:

1. We felt that the tremendous contributions of Ian Bernard Hickie, Jo-An Occhipinti, and Yun Ju Christine Song in supporting the writing and reviewing of the presented manuscript should be jointly recognized, providing merit in co-second authorship.

2. Thank you for your feedback. We have modified this sentence to say “…to support decision making, better managing systems challenges.”

 Reviewer #3:

Thank you for carefully addressing my comments. I think the paper has greatly improved. I just have a few minor issues.

1. The evaluation framework that you have developed includes a time dimension; some of the criteria refer to ex ante assessments (e.g. criterion 2), some to ex durante (e.g.criterion 3) and others ex post (e.g. e.g. criterion 25). I think it would improve the evaluation framework if this time dimension is included explicitly, so basically showing what questions need to be asked at what moment in time?

2. Not all criteria are clear. E.g. criterion 11: What are facilitators and barriers (e.g. inemtives)? facilitators and barriers for what?

3. p4-193 this sentence is unclear :" Stakeholders who have a ‘stake’ in the system, including those who are a part of the local context are viewed as important communication agents to decision makers, facilitating a positive movement whereby stakeholder involvement is perceived as almost necessary – shifting scientists away from working independently to develop systems models [21]."

4. p8-180: systematic reviews also use a protocol...?

5. p16-339: this sentence is unclear " There were two papers that did not explicitly consider the multi-value perspectives integrated within PM [54, 56].

6. p36-514: "In other words, knowledge that is acquired from evaluation outcomes can be applied prospectively throughout the PM process, rather than retrospectively reflecting on before-after measurements." Can you say a bit more about how your framework can be used pro-actively, for the design of PM processes? That would be useful!

Our response: Thank you for your thorough review.

1. Though we agree that the evaluation framework includes some reference to a time dimension, after careful consideration, we prefer to keep our evaluation framework as it is currently presented. We do explicitly mention the events around which the evaluation criteria can potentially be applied (p29-406), however, it is our intention that the evaluation framework be flexible and adaptive to participatory modelling programs of different structure and duration (as referenced in p7-173, p28-375, p34-475, p36-526, p38-561). 

2. Thank you for your feedback. We have carefully reviewed all evaluation criteria to ensure clarity and comprehension. 

3. We have rephrased the sentence (p4-193) to now read [changes denoted in CAPITAL LETTERS], “BY RECOGNIZING THE INTERDEPENDENCIES WITHIN COMPLEX SYSTEMS, DIVERSE STAKEHOLDER GROUPS, including those who are part of the SYSTEM BEING MODELLED are viewed as important change agents. THE INVOLVEMENT OF STAKEHOLDERS IS NECESSARY NOT ONLY FOR THEIR KNOWLEDGE CONTRIBUTIONS BUT ALSO FOR THEIR KEY ROLE IN COORDINATING THE IMPLEMENTATION OF STRATEGIC SYSTEM IMPROVEMENTS – HENCE THE VALUE IN SHIFTING SCIENTISTS AWAY FROM WORKING IN ISOLATION to develop systems models [21]."

4. To clarify, we do not say that systematic reviews do not use a protocol (p8-180). Rather, we have explicitly stated that traditional literature searches are not routinely informed by an a priori protocol. We have modified our sentence in our manuscript to make this distinction clearer to now read [changes denoted in CAPITAL LETTERS], “Scoping reviews also differ from NON-SYSTEMATIC literature searches as they are ROUTINELY informed by an a priori protocol, …” 

5. We have rephrased the sentence (p16-339) to now read [changes denoted in CAPITAL LETTERS], “There were two papers that did not explicitly consider the multi-value perspectives integrated within PM, SUCH AS THE CONSIDERATION OF THE MULTIPLE LEVELS OF EXAMINATION (E.G., DIVERSE STAKEHOLDER PERSPECTIVES, PROJECT VS SYSTEM LEVEL, SHORT VS LONG-TERM OBSERVATION, ETC) [54,56].”

6. We have added an additional sentence (p36-514) to now read [changes denoted in CAPITAL LETTERS]: "In other words, knowledge that is acquired from evaluation outcomes can be applied prospectively throughout the PM process, rather than retrospectively reflecting on before-after measurements. THROUGH PRINCIPLES OF PAR, THE PROPOSED EVALUATION FRAMEWORK EMBEDS CONTINUOUS CYCLES OF REFLECTION TO FACILITATE SHARED LEARNING AND ITERATIVE REFINEMENT OF PROCESSES THROUGHOUT PM EXAMINATION."

---

## [Decision Letter · Decision Letter 2]

15 Mar 2022

Presenting a comprehensive multi-scale evaluation framework for participatory modelling programs: A scoping review

PONE-D-21-25695R2

Dear Dr. Lee,

We’re pleased to inform you that your manuscript has been judged scientifically suitable for publication and will be formally accepted for publication once it meets all outstanding technical requirements.

Kind regards,

Fausto Cavallaro, PhD

Academic Editor

PLOS ONE

Reviewers' comments:

Reviewer's Responses to Questions

**Comments to the Author**

1. If the authors have adequately addressed your comments raised in a previous round of review and you feel that this manuscript is now acceptable for publication, you may indicate that here to bypass the “Comments to the Author” section, enter your conflict of interest statement in the “Confidential to Editor” section, and submit your "Accept" recommendation.

Reviewer #3: All comments have been addressed

2. Is the manuscript technically sound, and do the data support the conclusions?

Reviewer #3: Yes

3. Has the statistical analysis been performed appropriately and rigorously? 

Reviewer #3: N/A

4. Have the authors made all data underlying the findings in their manuscript fully available?

Reviewer #3: Yes

5. Is the manuscript presented in an intelligible fashion and written in standard English?

Reviewer #3: Yes

6. Review Comments to the Author

Reviewer #3: (No Response)

7. PLOS authors have the option to publish the peer review history of their article (what does this mean?). If published, this will include your full peer review and any attached files.

Reviewer #3: No

---

## [Editor Report · Acceptance letter]

31 Mar 2022

PONE-D-21-25695R2 

Presenting a comprehensive multi-scale evaluation framework for participatory modelling programs: A scoping review 

Dear Dr. Lee:

I'm pleased to inform you that your manuscript has been deemed suitable for publication in PLOS ONE. Congratulations! Your manuscript is now with our production department. 

Kind regards, 

on behalf of

Professor Fausto Cavallaro 

Academic Editor

PLOS ONE